# miR-205 mediates adaptive resistance to MET inhibition via ERRFI1 targeting and raised EGFR signaling

Cristina Migliore[1,2,*] , Elena Morando[2], Elena Ghiso[2], Sergio Anastasi[3], Vera P Leoni[4], Maria Apicella[2], Davide Cora'[1,2,5], Anna Sapino[2,6], Filippo Pietrantonio[7,8], Filippo De Braud[7,8], Amedeo Columbano[4], Oreste Segatto[3,**] & Silvia Giordano[1,2,***]

## Abstract

The onset of secondary resistance represents a major limitation to long-term efficacy of target therapies in cancer patients. Thus, the identification of mechanisms mediating secondary resistance is the key to the rational design of therapeutic strategies for resistant patients. MiRNA profiling combined with RNA-Seq in MET-addicted cancer cell lines led us to identify the miR-205/ERRFI1 (ERBB receptor feedback inhibitor-1) axis as a novel mediator of resistance to MET tyrosine kinase inhibitors (TKIs). In cells resistant to MET-TKIs, epigenetically induced miR-205 expression determined the downregulation of ERRFI1 which, in turn, caused EGFR activation, sustaining resistance to MET-TKIs. Anti-miR-205 transduction reverted crizotinib resistance *in vivo*, while miR-205 over-expression rendered wt cells refractory to TKI treatment. Importantly, in the absence of EGFR genetic alterations, miR-205/ERRFI1-driven EGFR activation rendered MET-TKI-resistant cells sensitive to combined MET/EGFR inhibition. As a proof of concept of the clinical relevance of this new mechanism of adaptive resistance, we report that a patient with a *MET*-amplified lung adenocarcinoma displayed deregulation of the miR-205/ERRFI1 axis in concomitance with onset of clinical resistance to anti-MET therapy.

**Keywords** EGFR; ERRFI1; MET; resistance; targeted therapy
**Subject Categories** Cancer; Pharmacology & Drug Discovery

## Introduction

MET is the receptor tyrosine kinase (RTK) for hepatocyte growth factor (HGF). MET triggering by HGF engagement activates a complex cellular program termed invasive growth, which protects from apoptosis and drives cells to proliferate and invade the surrounding tissues (Gherardi *et al*, 2012; De Silva *et al*, 2017). While physiologically required for tissue patterning during embryonic development and tissue homeostasis in post-natal life, the MET-driven invasive growth program can be also exploited by cancer cells in their quest to acquire growth autonomy and metastatic capabilities (Gherardi *et al*, 2012; De Silva *et al*, 2017).

MET over-expression linked to constitutive signaling is observed in several tumors, including gastric, lung, ovarian, renal, thyroid, liver, and esophageal cancers (Ghiso & Giordano, 2013). It may be caused by diverse mechanisms (hypoxia, activation of upstream genes, miRNA deregulation or *MET* gene amplification/exon 14 skipping mutations) and can confer to cancer cells a state of oncogene addiction. Accordingly, *MET* is categorized as a driver oncogene.

Among the different therapeutic approaches being exploited for suppressing MET oncogenic activity, selective (capmatinib, tepotinib) or non-selective (cabozantinib, crizotinib) small molecule tyrosine kinase inhibitors (TKIs) are in advanced clinical testing. Achieving a therapeutic success with MET-TKIs depends on patients' molecular selection, because only tumors truly addicted to MET signaling may respond to MET blockade. In addition, preclinical and clinical studies point to resistance as a vexing hurdle to the therapeutic success of MET-TKIs (Ghiso & Giordano, 2013). It follows that a detailed understanding of the signaling circuitries

1 Department of Oncology, University of Torino, Candiolo, Italy
2 Candiolo Cancer Institute, FPO-IRCCS, Candiolo, Italy
3 Unit of Oncogenomics and Epigenetics, IRCCS Regina Elena National Cancer Institute, Rome, Italy
4 Department of Biomedical Sciences, Unit of Oncology and Molecular Pathology, University of Cagliari, Cagliari, Italy
5 Department of Translational Medicine, Piemonte Orientale University "Amedeo Avogadro", Novara, Italy
6 Department of Medical Science, University of Torino, Torino, Italy
7 Medical Oncology, Fondazione IRCCS Istituto Nazionale dei Tumori, Milan, Italy
8 Department of Oncology and Hemato-oncology, University of Milano, Milan, Italy
*Corresponding author. Tel: +39 011993221; Fax: +39 011993225; E-mail: cristina.migliore@unito.it
**Corresponding author. Tel: +39 0652662551; Fax: +39 0652662600; E-mail: oreste.segatto@ifo.gov.it
***Corresponding author. Tel: +39 011993233; Fax: +39 011993225; E-mail: silvia.giordano@unito.it

underpinning resistance must be viewed as an integral component of the clinical development of MET-TKIs.

Several mechanisms of resistance to MET-TKIs have been already reported and include (i) *MET* amplification (Cepero *et al*, 2010), point mutations (Bahcall *et al*, 2016), and over-expression (Martin *et al*, 2014); (ii) *KRAS* amplification (Cepero *et al*, 2010); (iii) bypass activation of downstream pathways by RTKs acting in parallel to MET (Corso *et al*, 2010). Concerning the latter mechanism, altered miRNA expression is increasingly recognized as a strategy through which cancer cells reprogram their signaling circuitries in order to escape from pharmacological suppression of driver oncogenes (Migliore & Giordano, 2013). Herein, we report that miR-205 upregulation is sufficient to render MET-addicted tumors resistant to structurally different MET-TKIs (non-selective such as crizotinib, or selective such as PHA-665752 and JNJ-38877605) via *ERRFI1* targeting and consequent EGFR (epidermal growth factor receptor) activation. Accordingly, combined MET and EGFR pharmacological blockade reverts the adaptive resistance to MET-TKIs imposed by miR-205 upregulation.

## Results

### Generation and characterization of MET-TKI-resistant cell lines

GTL16 (gastric carcinoma cells), SG16 (primary gastric carcinoma cells), and EBC-1 (lung squamous cell carcinoma cells) are addicted to MET (Corso *et al*, 2010; Apicella *et al*, 2016). This phenotype is caused by *MET* amplification leading to MET over-expression and constitutive activation (Cepero *et al*, 2010; Apicella *et al*, 2016). In line, treatment of EBC-1, GTL16 and SG16 cells with crizotinib (non-selective MET-TKI) or PHA-665752 and JNJ-38877605 (selective MET-TKIs), strongly impaired their viability (Fig 1A–C).

We generated EBC-1, GTL16, and SG16 cells resistant to crizotinib, PHA-665752, or JNJ-38877605 by a stepwise dose escalation protocol, eventually obtaining derivatives capable of normal growth at drug concentrations roughly ten times higher than the $IC_{50}$ calculated for parental cells (Fig 1A–C). Notably, all the resistant cells were cross-resistant to the other MET-targeted drugs (Fig 1A–C).

To investigate a potential role for miRNAs in resistance to MET-TKIs, pairs of sensitive and resistant GTL16, EBC-1, and SG16 cells were profiled for miRNA expression. Out of 375 miRNAs examined, we found that 201, 98, and 140 miRNAs were expressed by GTL16, EBC-1, and SG16 cells, respectively. This compendium of expressed miRNAs was used for further analyses. Unsupervised hierarchical clustering of expressed miRNAs discriminated the three cell lines (Fig EV1A). When comparing relative levels of expressed miRNAs in each pair of sensitive/resistant cell lines, a single miRNA, namely miR-205, was concordantly and abundantly upregulated in all of the MET-TKI-resistant cells (Table 1). Real-time PCR analyses confirmed miR-205 upregulation (up to 130-fold) in resistant cells (Fig EV1B–D). This datum was consolidated by the observation that two additional MET-addicted cells lines, namely KATO II and SNU-5, showed increased miR-205 expression upon becoming resistant to MET-TKIs (Fig EV1E and F; and Appendix Fig S1A and B).

Since epigenetic regulation can modify miR-205 level (Hulf *et al*, 2013), we explored the possibility that differential methylation of *miR-205* genomic locus could contribute to the observed differences

of *miR-205* expression in resistant versus wt cells. To this aim, we investigated the methylation status of the *miR-205* genomic region (Appendix Fig S2A and B). As shown in Fig 1D, we observed that the level of methylation of the CpG enriched region mapping to the *miR-205* locus was significantly lower in resistant cells when compared to their wt counterpart. In fact, the CpG methylation level at the *miR-205* locus in resistant cells was comparable to that observed in parental cells upon treatment with 5-Aza-2′-deoxycytidine, with CpG de-methylation leading to increased miR-205 expression (Fig 1E).

### MiR-205 upregulation mediates resistance to MET-TKIs, which is linked to reduced expression of the putative miR-205 target ERRFI1

As shown in Fig 2A–C and Appendix Fig S3A–C, miR-205 silencing significantly reduced cell viability in all drug-resistant derivatives. Conversely, ectopic expression of miR-205 in wt cells was capable of increasing their viability at TKIs concentrations in the $IC_{50}$ range (Fig 2D–F and Appendix Fig S3D–F).

To further validate these *in vitro* results, we performed *in vivo* experiments. To this end, GTL16 R-CRIZ cells transduced with either control or anti-miR-205 lentivirus (Appendix Fig S4A) were injected s.c. in NOD-SCID mice. Tumor-bearing mice were subjected to treatment with crizotinib. As shown in Fig 2G, tumors generated by anti-miR-205 transduced cells were highly sensitive to crizotinib treatment, while controls remained resistant. In mirror experiments, tumors generated by wt GTL16 cells engineered to over-express miR-205 (Appendix Fig S4B) were refractory to crizotinib treatment, while control tumors were highly sensitive (Fig 2H).

The above dataset is compatible with the hypothesis that miR-205 upregulation can cause MET-addicted cancer cells to acquire resistance to MET-TKIs. This, in turn, raises the question of which among miR-205 targets may determine the observed resistance phenotype. In broad terms, resistance to TKIs may be caused by mutations of the target kinase, resulting in reduced/abolished TKI binding, or bypass activation of downstream pathway(s) despite enduring target blockade (Lackner *et al*, 2012). In general, MET kinase retained sensitivity to inhibition by MET-TKIs in resistant cells even if this was not complete. Treatment with MET-TKIs did not translate into significant suppression of ERK and AKT activation in resistant cells, although in CRIZ-resistant GTL16 cells AKT activation was not as high as in the wt control (Fig 2I). Of note, activation of ERK and AKT was affected marginally, or not at all, by drug withdrawal (Fig 2I), suggesting that preservation of oncogenic signaling downstream to MET was not a direct effect of MET-TKI administration.

In searching for mechanisms responsible for bypass signal activation, we found that expression of PTEN, a bona fide miR-205 target gene (Cai *et al*, 2013), was not altered in resistant cells (Appendix Fig S5), while EGFR, a well-known MET partner (Haura & Smith, 2013), retained or even increased its expression/activity in resistant cells (Fig 2I). No *EGFR* mutation/amplification was detected in resistant cells. These data suggest a potential role for non-mutational EGFR activation in fueling vicarious ERK and AKT activation in MET-TKI-resistant derivatives subjected to MET blockade.

A potential link between miR-205 upregulation and increased EGFR activity emerged from the comparative analysis of RNA-Seq

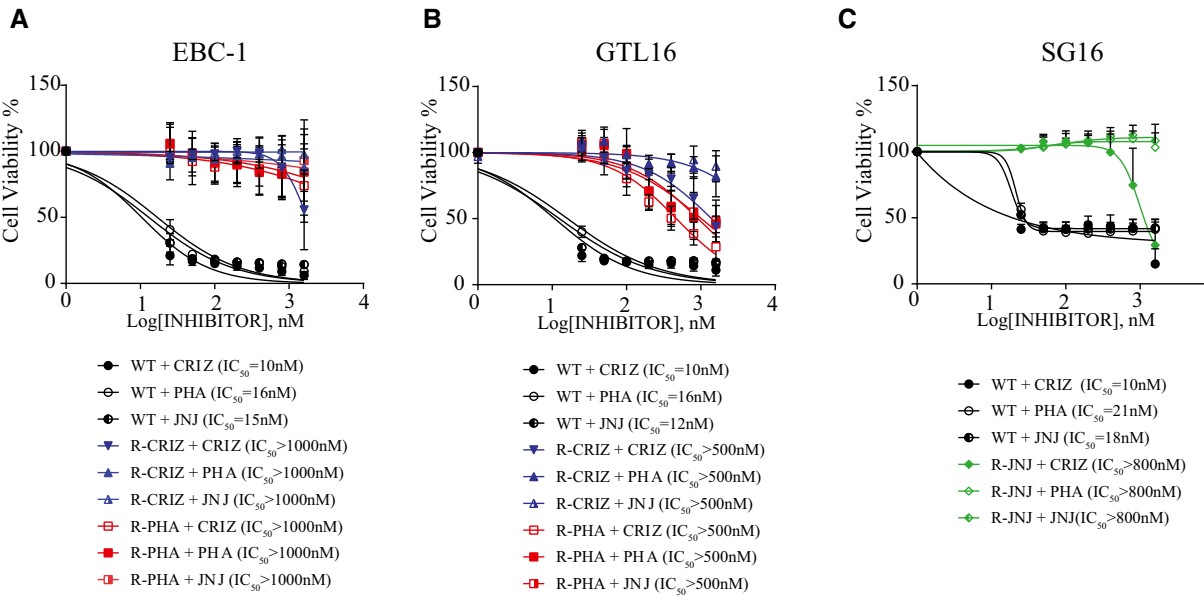

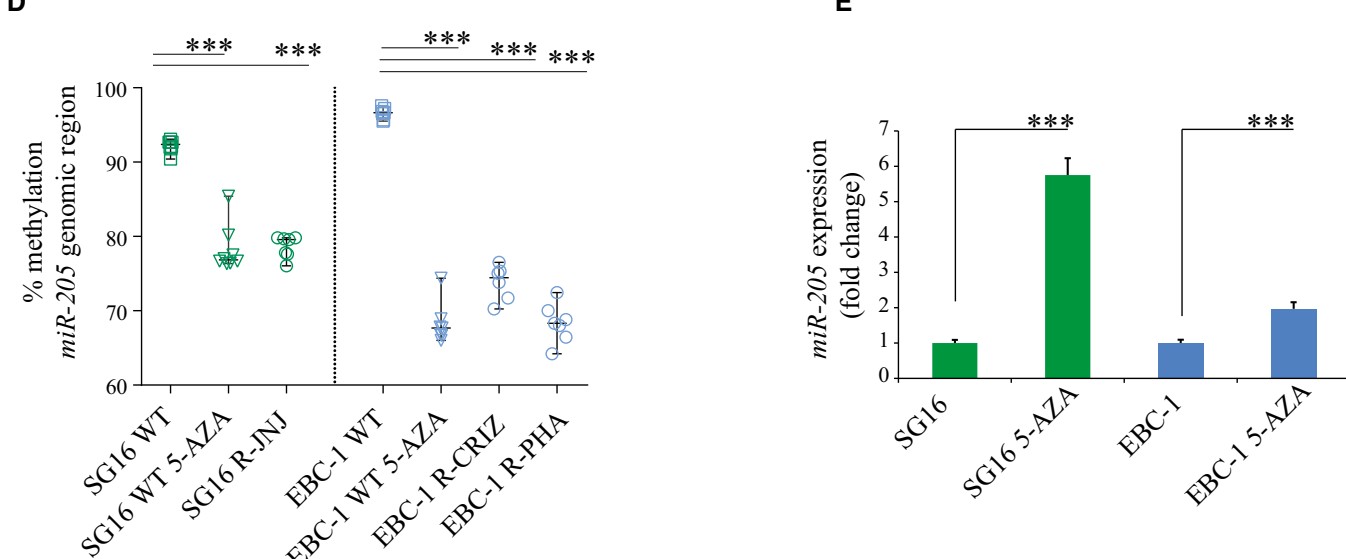

**Figure 1. Characterization of MET-addicted cells rendered resistant to MET tyrosine kinase inhibitors.**

A–C EBC-1 (A), GTL16 (B), and SG16 (C), either parental (wt) or resistant (-R) to the indicated MET-TKIs, were exposed to escalating concentrations of the indicated drugs. Cell viability was measured after 72 h (CellTiterGlo) to derive IC$_{50}$ values. Data are the mean of three independent experiments, each performed in quadruplicate wells. PHA = PHA-665752; CRIZ = crizotinib; JNJ = JNJ-38877605; R-PHA = cells resistant to PHA-665752; R-CRIZ = cells resistant to crizotinib; R-JNJ = cells resistant to JNJ-38877605.

D SG16 and EBC-1 genomic DNA from resistant and wt cells, untreated or treated with 5-Aza-2′-deoxycytidine (5-AZA), was extracted and subjected to bisulfite conversion. The *miR-205* genomic region was amplified by PCR and pyrosequenced. The scatter dot plot represents the percentage of DNA methylation of *miR-205* genomic region; each dot exemplifies the average of the methylation status of six CpGs (shown in Appendix Fig S2) analyzed in two technical replicates (Squares = wt untreated cells; triangles = wt cells treated with 5-AZA (used as control); circles = resistant cells).

E *MiR-205* expression was evaluated by RT–qPCR in SG16 and EBC-1 wt treated or not with 5-Aza-2′-deoxycytidine (5-AZA). As shown, *miR-205* level significantly increased upon 5-AZA treatment. *n* = 3 per condition.

Data information: (A–C) Data are presented as mean ± SD. Two-way ANOVA, Bonferroni's multiple comparisons test was significant (*P* < 0.001) in wt versus resistant cells. (D) Data are presented as median and 95% CI. ***P* < 0.001; one-way-ANOVA, Bonferroni's multiple comparisons test. (E) Data are presented as mean ± SD. ****P* < 0.001, two-tailed *t*-test.

**Table 1.  Global miRNA expression measured by TaqMan low density array.**

| | EBC1 R150 CRIZ | | EBC1 R250 PHA | | GTL16 R200 CRIZ | | GTL16 R300 PHA | | SG16 R250JNJ | |
|---|---|---|---|---|---|---|---|---|---|---|
| | miRNA | Fold change | miRNA | Fold change | miRNA | Fold change | miRNA | Fold change | miRNA | Fold change |
| TOP 10 miRNAs UPREGULATED | miR-205 | 53.1 | miR-146a | 1301.5 | miR-205 | 1256.9 | miR-205 | 1368.7 | miR-339-3p | 8.1 |
| | miR-27b | 1.8 | miR-205 | 115.4 | miR-328 | 528.7 | miR-125b | 305.6 | miR-339-5p | 7.7 |
| | miR-27a | 1.5 | miR-27b | 7.7 | miR-133a | 330.0 | miR-501-3p | 260.6 | miR-148a | 7.5 |
| | miR-193b | 1.4 | let-7a | 4.6 | miR-146a | 245.8 | miR-184 | 104.1 | miR-375 | 6.9 |
| | miR-141 | 1.4 | miR-422a | 3.0 | miR-125b | 179.5 | miR-146a | 81.5 | miR-205 | 5.8 |
| | miR-24 | 1.4 | miR-146b-5p | 2.7 | miR-642 | 94.6 | miR-99a | 58.1 | miR-328 | 5.4 |
| | let-7a | 1.2 | miR-24 | 2.7 | miR-423-5p | 93.9 | miR-495 | 46.9 | miR-9 | 5.3 |
| | miR-19a | 1.2 | miR-27a | 2.3 | miR-579 | 91.8 | miR-133a | 41.0 | miR-10a | 5.2 |
| | miR-9 | 1.1 | miR-598 | 2.3 | miR-125a-3p | 50.4 | miR-616 | 37.3 | miR-139-5p | 4.9 |
| | miR-365 | 1.1 | miR-200a | 2.3 | miR-142-3p | 48.8 | miR-15a | 27.6 | miR-23b | 4.7 |
| TOP 10 miRNAs DOWNREGULATED | miR-196b | 0.1 | miR-100 | 0.1 | miR-363 | 0.0 | miR-450b-5p | 0.0 | miR-106b | 0.3 |
| | miR-125b | 0.1 | miR-99a | 0.1 | miR-379 | 0.2 | miR-363 | 0.0 | miR-155 | 0.4 |
| | miR-146a | 0.1 | miR-125b | 0.1 | miR-517c | 0.2 | miR-22 | 0.1 | miR-93 | 0.4 |
| | miR-100 | 0.2 | miR-886-5p | 0.1 | miR-410 | 0.2 | miR-190 | 0.1 | miR-25 | 0.5 |
| | miR-99a | 0.2 | miR-139-5p | 0.2 | miR-518f | 0.4 | miR-517c | 0.1 | miR-100 | 0.6 |
| | miR-345 | 0.2 | miR-138 | 0.2 | miR-190 | 0.4 | miR-150 | 0.2 | miR-99a | 0.7 |
| | miR-886-3p | 0.2 | miR-886-3p | 0.3 | miR-20b | 0.4 | miR-452 | 0.2 | miR-345 | 0.7 |
| | miR-339-5p | 0.3 | miR-222 | 0.4 | miR-505 | 0.5 | miR-449b | 0.3 | miR-10b | 0.7 |
| | miR-212 | 0.3 | has-miR-155 | 0.4 | miR-9 | 0.5 | miR-362-3p | 0.3 | miR-320 | 0.7 |
| | miR-138 | 0.3 | miR-484 | 0.4 | miR-376a | 0.5 | miR-338-3p | 0.4 | miR-18a | 0.7 |

data obtained from wt and crizotinib-resistant pairs of both EBC-1 and GTL16 cells. Figure 2J and Appendix Fig S6 show that *ERRFI1* (ERBB receptor feedback inhibitor 1) was one of the few putative miR-205 targets, as predicted by the TargetScan algorithm (Agarwal *et al*, 2015), to be downregulated in both EBC-1- and GTL16-resistant cells (Cora' *et al*, 2017). The *ERRFI1* product (also named MIG6) is an inducible feedback inhibitor of the EGFR/HER receptor family (Anastasi *et al*, 2016). Genetic studies in the mouse have pointed to an essential role of Errfi1 in restraining Egfr-dependent cell proliferation in normal tissues as well as suppressing Egfr-driven tumor formation (Anastasi *et al*, 2016). Mechanistically, ERRFI1 binds to the EGFR activated kinase domain, thus suppressing its catalytic activity. In addition, ERRFI1 instigates endocytosis/degradation of the kinase-inactive EGFR molecules to which it binds (Frosi *et al*, 2010). Hence, we surmised that ERRFI1 downregulation consequent to miR-205 uprise could mediate an increase in EGFR expression/activity (see Fig 2I) sufficient to fuel resistance to MET blockade. In line, ERRFI1 expression was clearly lower in resistant cells compared to their parental counterpart (Fig EV2A–C). GTL16 R-PHA cells stood out as the single exception, most likely because *KRAS* amplification mediates the resistance of these cells to PHA-665752 (Cepero *et al*, 2010). Ectopic ERRFI1 sufficed to re-sensitize resistant cells to MET-TKIs both *in vitro* (Fig 3A and B; and Appendix Fig S7A and B) and *in vivo* (Fig EV3A and B), with the predictable exception of GTL16 R-PHA cells (Appendix Fig S7B). Critically, ERRFI1 knockdown attenuated growth suppression of

parental EBC-1 and SG16 cells mediated by either crizotinib or JNJ-38877605 (Fig 3C). We noted that ectopic ERRFI1 was not as effective as anti-miR-205 in restoring sensitivity to crizotinib in tumor xenotransplants (compare Fig 2G with Fig EV3B). The most parsimonious explanation for this discrepancy is that while *ERRFI1* appears to be the most critical miR-205 target in this context, other miRNA targets may contribute to the development of the resistant phenotype. The above data provide a causal link between loss of ERRFI expression and the development of resistance to MET-TKIs in MET-addicted cells. Interestingly, we observed that the miR-205/ERRFI1/EGFR regulatory axis can operate also in not-addicted cells, as ectopic miR-205 expression in A549 lung cancer cells resulted in ERRFI1 downregulation and increased EGFR expression/activation (Appendix Fig S8A and B).

### ERRFI1 targeting by miR-205 increases EGFR activity, promoting resistance to MET-TKIs

To investigate whether miR-205 is involved in *ERRFI1* regulation, we expressed ectopic miR-205 in EBC-1, GTL16, and SG16 wt cells. This was sufficient to produce a decrease in ERRFI1 protein expression (Fig 3D). In a second set of experiments, we investigated the regulation of a Luc-ERRFI1 reporter (in which the *ERRFI1* 3′ UTR is cloned downstream to the luciferase cDNA) in wt GTL16 cells that express very low levels of endogenous miR-205. Figure 3E shows that ectopic miR-205 significantly downregulated the luciferase

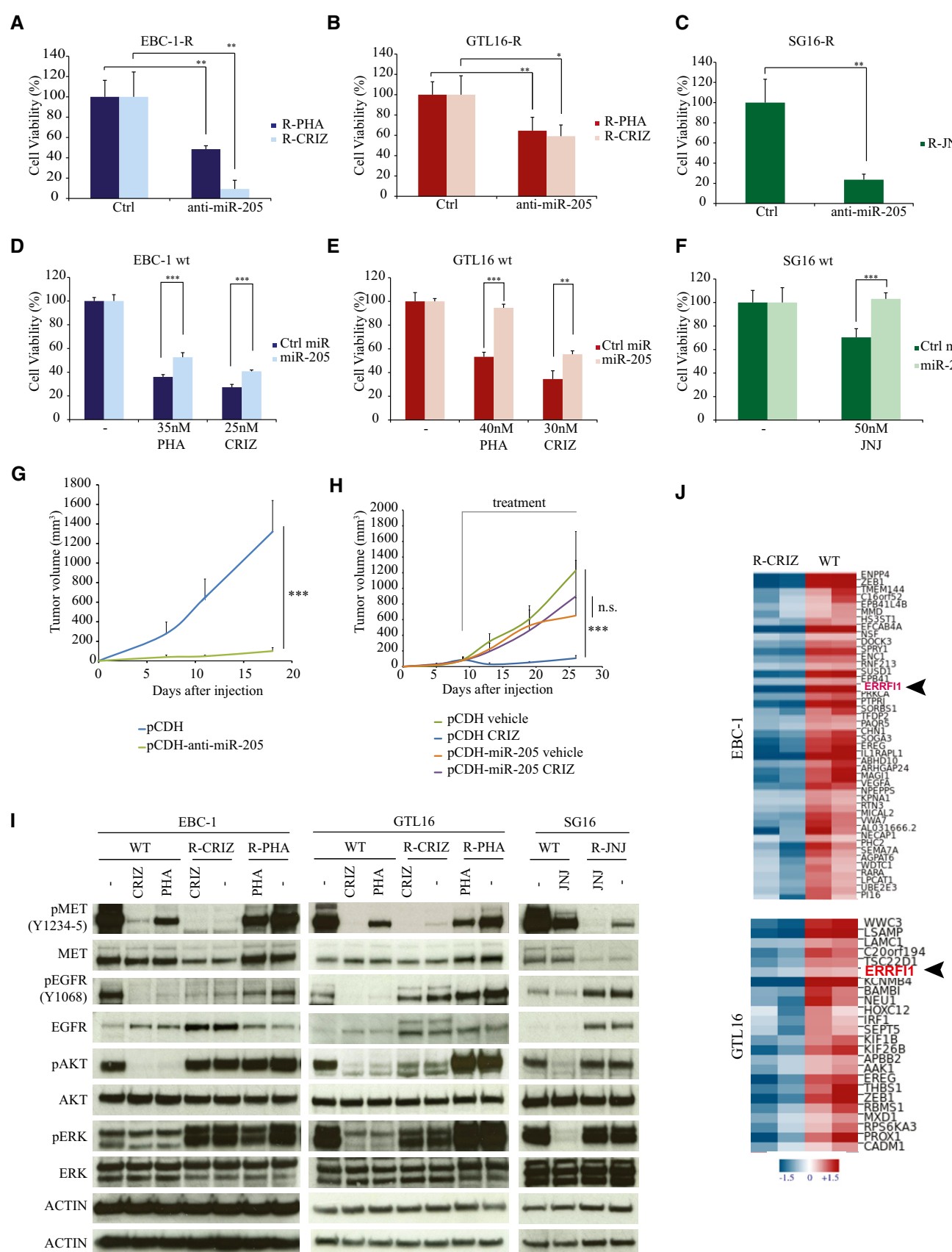

**Figure 2.**

**Figure 2.  miR-205 modulates tumor cell sensitivity to MET-TKIs.**

A–C    *Mir-205* expression was silenced by transfection of either anti-miR-205 or control antagomiR (Ctrl) in EBC-1 (A), GTL16 (B), and SG16 (C) resistant cells. Resistant cells were grown in the presence of the TKI to which they are resistant; viability was assessed after 72 h by CellTiterGlo. *n* = 4 per condition.

D–F    MiR-205 or a control miRNA (Ctrl miR, a random miRNA sequence) was over-expressed in parental (wt) EBC-1 (D), GTL16 (E), and SG16 (F) cells. Cells were grown for 72 h in the presence of MET-TKIs at the indicated doses. Viability was evaluated as above. *n* = 4 per condition.

G    GTL16 R-CRIZ cells transduced with pCDH-anti-miR-205 or pCDH (control vector) were subcutaneously injected in NOD/SCID mice. Mice were treated with crizotinib (25 mg/kg), and tumor volume was monitored for 18 days as indicated. *n* = 6 per condition.

H    GTL16 wt cells transduced with pCDH-miR-205 or with the control vector (pCDH) were injected in NOD/SCID mice. When tumors reached an average volume of around 150 mm³, treatment with either crizotinib (12.5 mg/kg) or vehicle was started. Tumor volume was monitored for 18 days as indicated. *n* = 6 per condition.

I    Western blot analysis of EBC-1, GTL16, and SG16 cells, either parental (wt, untreated or treated for 2 h with the indicated TKIs) or TKI-resistant (either in the presence or absence of the TKIs to which they are resistant). Cell lysates were probed with the indicated antibodies. Actin was used as loading control (one actin panel for each WB performed/cell line). Drug abbreviation is as shown in Fig 1.

J    The transcriptome of crizotinib-sensitive/resistant pairs of GTL16 and EBC-1 cells was determined by RNA-Seq. The mRNA heatmap shows the expression levels of the predicted miR-205 targets (TargetScan 7.1) downregulated in resistant versus wt cells. For each gene, the log2 of the RSEM expected counts was converted to the log2 ratio against the global median expression of the gene in all samples. Log2 ratio values were loaded in GEDAS software to perform hierarchical clustering analysis and represent data in a heatmap. Units (−1.5 to +1.5) represent the log2 ratio against the median. Red and blue colors represent the highest and lowest ends of mRNA expression levels, respectively.

Data information: (A–H) Data are presented as mean ± SD. Asterisks: ***P < 0.001; **P < 0.01; *P < 0.05. Two-tailed *t*-test was used for panels (A–C, F); two-way ANOVA, Bonferroni's multiple comparisons test was used for panels (D, E, G, H).
Source data are available online for this figure.

---

signal in Luc-ERRFI1-transfected cells, an effect comparable to that observed upon expression of miR-200c, a validated *ERRFI1*-targeting miRNA (Adam *et al*, 2009). Deletion or mutation of the predicted miR-205 targeting sequence in the Luc-*ERRFI1* reporter strongly reduced luciferase downregulation by ectopic miR-205 (Fig 3F), pointing to direct regulation of *ERRFI1* by miR-205. Finally, and in line with ERRFI1 acting as EGFR inhibitor, ectopic miR-205 markedly increased EGFR expression/activation in EBC-1 and SG16 cells (Fig 3G). A more modest effect was observed in GTL16 cells.

Increased EGFR activity, consequent to miR-205-dependent downregulation of ERRFI1, was largely responsible for resistance to MET-TKIs, because the clinically approved EGFR-TKI afatinib greatly alleviated resistance of EBC-1 and SG16 cells to MET-TKIs in a dose-dependent manner (Fig 3H and I). A less pronounced effect was observed also in GTL16 cells resistant to crizotinib (Appendix Fig S9). Overall, the activity of afatinib in individual resistant cell lines correlated with the magnitude of relative changes in cell proliferation rates observed upon miR-205 downregulation (Fig 2A–C); afatinib activity in resistant cells also correlated with

relative variations of pEGFR levels detected in the corresponding parental cells upon miR-205 over-expression (Fig 3G). Finally, GTL16 R-PHA growth was not affected by afatinib, consistent with *KRAS* amplification being causative of resistance to PHA-665752 (Appendix Fig S9).

## Concurrent ERRFI1 downregulation and miR-205 over-expression are identified in a patient with acquired MET-TKI resistance

Motivated by the above findings, we endeavored to investigate whether alterations of the miR-205-*ERRFI1* axis can enforce clinical resistance to MET-TKIs. Because MET-TKIs are not yet approved for routine clinical use, we could not have access to specimens from patients harboring MET-driven tumors and undergoing treatment with MET-TKIs. However, we could study two cases—a *BRAF*-mutated CRC (patient #1) and an *EGFR*-mutated NSCLC (patient #2)—with documented MET-driven acquired resistance to the respective target therapy. Of note, administration of a MET-TKI led in both cases to a partial response followed by relapse.

---

**Figure 3.  ERRFI1 is the main miR-205 target responsible for resistance to MET-TKIs.**

A, B    EBC-1 (A) and SG16 (B) resistant cells were transduced with either empty (pCDH) or ERRFI1-encoding (pCDH ERRFI1) recombinant lentivirus stocks. Resistant cells (R-CRIZ, R-PHA, R-JNJ) were grown in the presence of the TKIs to which they are resistant. As control, wt cells were grown in the absence (NT) or in the presence of the indicated MET-TKIs. Viability was assessed by CellTiterGlo 72 h after plating. *n* = 4 per condition.

C    ERRFI1 expression was silenced in EBC-1 and SG16 wt cells by transfection of either control (siCtrl) or *ERRFI1*-targeting siRNA pools. Cells were treated with MET-TKIs at concentrations averaging the respective IC$_{50}$; viability was evaluated 72 h later. *n* = 4 per condition.

D    Immunoblot analysis of ERRFI1 in cell lines expressing ectopic miR-205. Actin was used as total protein loading control. The column chart shows the ratio between the ERRFI1 and actin band intensities as quantified by ImageJ software.

E    GTL16 wt cells were co-transfected with the Luc-ERRFI1 reporter plasmid along with the indicated miRNAs. Data are computed from six independent experiments and expressed as arbitrary units (RLU), one being the control value obtained in cells expressing a non-targeting miRNA. *n* = 6 per condition.

F    GTL16 wt cells were co-transfected with miR-205 and the Luc-ERFFI1 reporter, either in wt or delta, or mutated configuration. Data are computed from three independent experiments and expressed as in (E). *n* = 3 per condition.

G    Western blot analysis of EGFR expression and activation (pEGFR) in wt EBC-1, GTL16, and SG16 cells transfected with a control miRNA (Ctrl) or miR-205. Actin was used as total protein loading control. Column chart shows the ratio between the pEGFR and actin band intensities as quantified by ImageJ software.

H, I    Viability of EBC-1 (H) and SG16 (I) cells, either wt or resistant to indicated MET-TKIs, following a 72 h incubation with escalating concentrations of afatinib. Resistant derivatives were co-treated with the MET-TKI to which they are resistant. *n* = 4 per condition.

Data information: (A–C, E, F, H, I) Data are presented as mean ± SD. ***P < 0.001; **P < 0.01; *P < 0.05, n.s. not significant. Two-tailed *t*-test (A–C, F), one-way ANOVA (E), two-way ANOVA, Bonferroni's multiple comparisons (H, I).
Source data are available online for this figure.

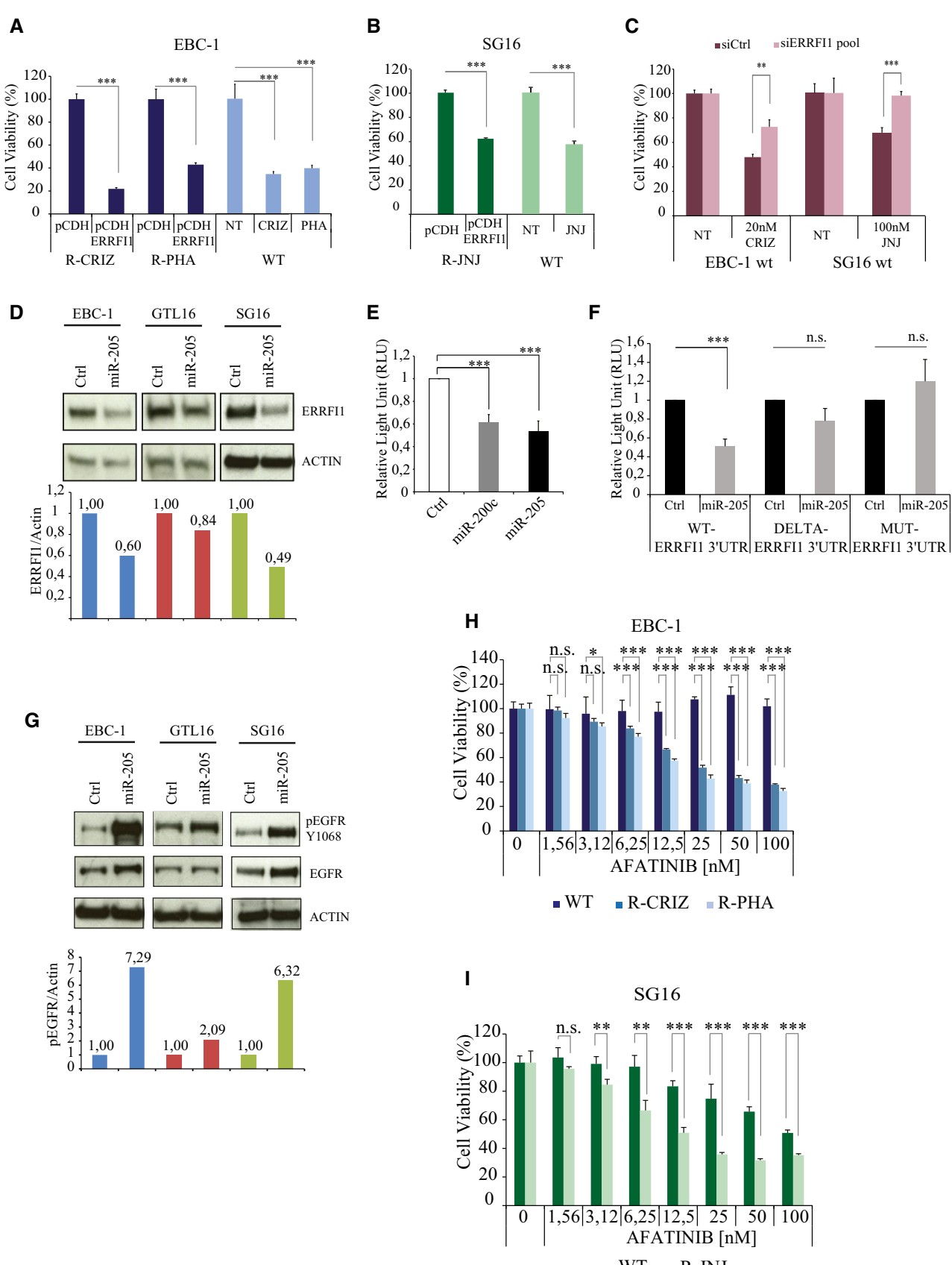

**Figure 3.**

In patient #1 (whose clinical history is reported in Pietrantonio *et al*, 2016; Oddo *et al*, 2017), IHC did not reveal significant differences in ERRFI1 levels when acquired resistance to MET plus BRAF inhibition became manifest (Fig EV4). Interestingly, *MET* hyper-amplification was reported as causative of resistance to MET blockade in this patient (Oddo *et al*, 2017).

In patient #2, whose clinical history is summarized in Fig EV5, IHC analysis showed that ERRFI1 expression was decreased in cells that became resistant to dual MET/EGFR blockade compared to cells resistant to single agent erlotinib [compare panel (b) with panel (a) in Fig 4A]. Notably, ERRFI1 levels in (a) and (b) were anti-correlated to coincident changes in *miR-205* expression (Fig 4B). Because it occurred in the context of ongoing gefitinib administration (Fig EV5), we assume that loss of ERRFI1 expression was sufficient to enhance signaling by a fraction of EGFR del746-750 escaping gefitinib blockade. This is consistent with the notion that *Errfi1* is haploinsufficient in a mouse model of EGFR del746-750-driven NSCLC (Anastasi *et al*, 2016). The above data provide a proof of concept that anti-correlated oscillations of miR-205 and *ERRFI1* expression do occur in a clinical context of resistance to a MET-TKI, predictably enforcing an EGFR-dependent mechanism of refractoriness to MET blockade.

## Discussion

Several MET-TKIs are in advanced clinical development; however, as in the case of TKIs targeting other oncogenic RTKs, the emergence of resistance poses a serious challenge to achieving a long-term benefit from MET-TKIs (Haura & Smith, 2013). Here, we describe a novel mechanism of adaptive resistance to three structurally different MET-TKIs, which entails upregulation of EGFR activity consequent to miR-205-driven reduction in ERRFI1 expression. This is yet another example of how versatile and effective miRNA regulation might be in shaping the adaptation of cancer cells to external perturbations. As for mechanism/s involved in miR-205 upregulation, our initial studies suggest a role for de-methylation of regulatory sequences in the *miR-205* genomic locus.

Our demonstration that miR-205-driven ERRFI1 loss is sufficient to confer vicarious oncogenic properties to EGFR signaling is congruent with (i) biochemical studies that assign to ERRFI1 a role as pan-ERBB inhibitor; (ii) genetic analyses in the mouse that have nominated Errfi1 as a tumor suppressor; (iii) biological experiments in glioblastoma, NSCLC, and pancreatic carcinoma cells that point to ERRFI1 loss as key mechanism in sustaining oncogenic addiction to EGFR signaling (reviewed in Anastasi *et al*, 2016). Interestingly, we did not observe a consistent miR-205 increase in

**A**

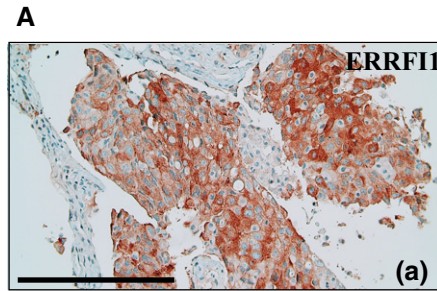
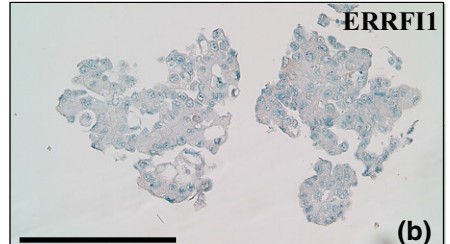

**B**

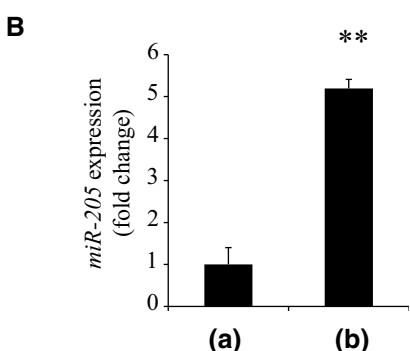

**Figure 4. Concomitant ERRFI1 downregulation and miR-205 over-expression are identified in a patient with acquired resistance to MET-TKIs.**

A   Immunohistochemical analysis of ERRFI1 expression in biopsies taken from patient #2 upon onset of resistance to EGFR therapy (a) and, subsequently, to combined anti-EGFR/MET treatment (b). Note that (a) and (b) refer to the stages of patient's clinical history outlined in Fig EV5. Scale bar = 0.25 mm.

B   MiR-205 expression was evaluated by RT–qPCR on FFPE-derived RNA extracted from samples (a) and (b). *n* = 3, average ± SD. \*\**P* < 0.01, two-tailed *t*-test.

EGFR-addicted cells rendered resistant to different EGFR inhibitors (Appendix Fig S10); this suggests that miR-205 upregulation might operate selectively in rewiring EGFR activity in the context of bypass resistance to non-EGFR TKIs.

Remarkably, the above mechanism of resistance to MET-TKIs, albeit not genetically driven, is therapeutically actionable, because EGFR blockade suffices to restore sensitivity to MET-TKIs. Although the clinical use of EGFR TKIs is essentially restricted to *EGFR*-mutated NSCLC, our data imply that selected contexts of clinical resistance to MET-TKIs may represent a prime example of beneficial targeting of wt *EGFR*. We duly caution that further work on a larger scale is needed to validate our model of deregulated miR-205-ERRFI1-EGFR axis in MET-addicted tumors. We envision that pathological samples obtained from patients enrolled in ongoing clinical studies (e.g., the NCT02414139 trial that is evaluating the efficacy of the MET-TKI capmatinib in NSCLC patients selected for the presence of MET deregulation) will be invaluable for future studies.

The present study expands on previous work by the Sidranski's group (Izumchenko *et al*, 2014), who showed that *ERRFI1* targeting by miR-200 (a known *ERRFI1* targeting miRNA) heightens EGFR signaling and consequently generates a condition of EGFR addiction in lung and pancreas cancer cells expressing wt *EGFR*. We hypothesize that the dominant role of a given miRNA in *ERRFI1* regulation may reflect variables such as tumor cell identity, stage of tumor evolution, and therapeutic setting. Previous work also pointed to miR-200 and *ERRFI* as reliable surrogate markers of addiction to wt EGFR (Anastasi *et al*, 2016). This is notable, as routine evaluation in the clinical laboratory of EGFR Tyr-phosphorylation is still technically challenging. In the setting of MET-addicted cancers, our work anticipates that assessing ERRFI1 and miR-205 expression could guide the identification of patients that develop adaptive EGFR-driven resistance to MET-TKIs.

*MET* amplification occurs in up to 18% of *EGFR*-mutated NSCLC cases that develop resistance to approved EGFR TKIs (Engelman & Jänne, 2008). Previous work (Apicella *et al*, 2016), along with preclinical and clinical data presented herein, supports the view that EGFR activation can in turn substitute for MET signaling in MET-driven cancer cells. Thus, MET and EGFR appear to be bound by a mutual rescue agreement whenever any of the two incurs in pharmacological blockade. In line with this proposition and based on promising proof-of-concept studies (Corso *et al*, 2010; Haura & Smith, 2013), the addition of MET-TKIs to EGFR inhibitors is currently under clinical investigation in the setting of MET-driven acquired resistance to EGFR blockade. More in general, our data suggest that upfront co-targeting of MET and EGFR could be a viable option to delay/prevent/revert the onset of TKI resistance in at least a fraction of MET- or EGFR-addicted tumors. While the identification of robust biomarkers capable of identifying tumor subsets that may benefit from wt EGFR targeting is a major challenge ahead, the present work provides a mechanism-based rationale for future clinical and translational research.

# Materials and Methods

## Patient care and specimen collection

Clinical data of patient #1 were reported in references (Pietrantonio *et al*, 2016; Oddo *et al*, 2017).

At diagnosis, patient #2 was a 52-year-old male with metastatic *EGFR* exon 19-deleted lung adenocarcinoma. Tumor samples were collected in accordance with an Institutional Review Board-approved protocol, to which the patient provided written informed consent. Informed consent was obtained from all subjects. The experiments conformed to the principles set out in the WMA Declaration of Helsinki and the Department of Health and Human Services Belmont Report. Initial therapy with erlotinib was given as per standard practice, whereas gefitinib/capmatinib was subsequently administered as experimental combination (clinicaltrials.gov identifier: NCT01610336).

## Animal studies

Animal handling and experimentation was performed in accordance with the European Union directives and the Italian Guidelines for Care and Use of Laboratory Animals. All animal procedures were approved by the Ethics Committee of the Candiolo IRCC and the Italian Ministry of Health. All experiments were performed in 7- to 8-week-old female NOD-SCID mice purchased from Charles River (Milan, Italy) and maintained on a 12 h light/dark cycle at 22°C. In each animal experiment, mice were randomly assigned to each group.

$1 \times 10^6$ wt or R-CRIZ GTL16 cells were injected subcutaneously into the right posterior flanks of 6-week-old immunodeficient NOD/SCID female mice. Mice carrying tumors from wt GTL16 cells were randomized and treated or not with crizotinib (12.5 mg/kg) when tumors reached a volume of around 150 mm$^3$, while for animals injected with R-CRIZ GTL16 cells crizotinib administration (25 mg/kg) started from the beginning. Tumor size was evaluated by caliper measurements, and approximate volume of the mass was calculated using the formula $4/3 \pi (D/2) (d/2)2$, where d and D are the minor tumor axis and major tumor axis, respectively. Crizotinib (PF-02341066) was obtained from the Candiolo IRCCs Pharmacy that collected small amount of drugs left over after treating patients. Experiments were not performed in blind.

## Cell cultures, plasmids, and compounds

HEK293T, A549, and SNU-5 cell lines were obtained from ATCC (Manassas, VA, USA); EBC-1 from Japan Cancer Research Resources Bank (Ibaraki, Osaka, Japan); KATO II cells were kindly provided by Dr. Yitzhak Zimmer, University of Bern. GTL16 and SG16 were described in references (Migliore *et al*, 2008; Apicella *et al*, 2016). The genetic identity of the cell lines was periodically checked by short tandem repeat profiling (Cell ID, Promega, Madison, Wisconsin, USA). *Mycoplasma* testing is performed routinely using the PCR Mycoplasma Detection Kit (Applied Biological Materials Inc., Richmond, BC, Canada).

PHA-665752 (PHA) and afatinib were from Selleckchem (Munich, Germany); crizotinib (CRIZ) from Sequoia Research Products (Pangbourne, United Kingdom). The MET inhibitor JNJ-38877605 (JNJ) was provided by Janssen Pharmaceutica NV (Beerse, Belgium). To generate resistant cell lines, we used a stepwise dose escalation method, starting from a dose close to the cell viability IC$_{50}$ followed by stepwise escalations over a 6- to 12-month period. Established resistant sublines were maintained in culture with the highest drug dose compatible with full viability and

proliferation rates similar to those of parental cells. All the assays involving resistant cells were performed in the presence of the TKI to which they had been rendered resistant, i.e., at the maximum dose reached at the end of the dose escalation protocol.

The ERRFI1 expression vector was described elsewhere (Frosi et al, 2010). Lentiviral construct for anti-miR-205 was kindly provided by Dr. Chun-Ju Chang, Purdue University. Luciferase reporter vector was obtained cloning the full ERRFI1 3′UTR sequence (nt. 1639-3144) downstream to the luciferase gene in pMIR-REPORT vector (Thermo Fisher, Waltham, MA, USA). The delta ERRFI1 3′UTR vector was produced by deleting nt 2923–3144 of the ERRFI1 3′UTR, containing the unique miR-205-5p target sequence, by single digestion with PmeI. MUT-ERRFI1-3′UTR vector was produced as follows. A SacI-NotI fragment encompassing the ERRFI1 3′UTR sequence was excided from pMIR-Report-ERRFI1 and cloned into the pBluescript (BS) plasmid. The resulting vector, pBS-hERRFI1 3′UTR, was used as a template for mutagenesis using the Q5 site-directed mutagenesis kit (New England Biolabs, NEB). The mutagenized insert was transferred back into pMIR-Report. The following primers, designed according to the NEB software NEBase Changer (https://nebasechanger.neb.com), were used in the mutagenesis reaction:

Forward Primer: 5′-TATGAACTAAccGcgaGTTAAAACATGCTTAAGAAAAATGCAC-3′
Reverse Primer: 5′-CTAAAATATAATAAGCTTTAAATAGC-3′

In detail, the ATGAAGG sequence (corresponding to position 2,980–2,986 of the ERRFI1 cDNA sequence) was mutagenized to ccGcgaG (base changes in lower case). Mutagenesis and sub-cloning steps were sequence verified.

## Protein extraction and Western blot

For Western blot analysis, cells were lysed in LB buffer [2% SDS, 0.5 mol/l Tris–HCl (pH 6.8)]. Western blots were performed according to standard methods. Primary antibodies were as follows: anti-phospho-MET#3077 (Y1234-5); anti-AKT#9272; anti-phospho-AKT#4060 (S473); anti-MAPK#9102; anti-phospho-MAPK#9101 (Thr202/Tyr204); anti-PTEN#9188 from Cell Signaling (Leiden, The Netherlands); anti-β-actin #A3854 from Sigma; anti-phospho-EGFR#5644 (Y1068) from Abcam (Cambridge, UK); anti-EGFR (sc71033); anti-myc (sc-40B) and anti-HER2 (sc284) from Santa Cruz; and anti-ERRFI1 was described in Frosi et al (2010). All antibodies were used at a dilution of 1:1,000, except for anti-actin (1:50,000) and anti-ERRFI1 (1:2,500). TKIs were added 2 h before cell lysis.

## Gene transfer procedures and cell viability assay

For cell viability assays, cells were seeded in quadruplicate well in 96-well culture plates ($3–5 \times 10^3$ cells/well), in the presence of the indicated drugs. After 72 h, cell viability was measured using the Cell Titer-Glo Luminescent Cell Viability Assay (Promega).

EBC-1, GTL16, and SG16 cells were transfected with siRNAs/miRNAs using Lipofectamine 2000 (Thermo Fisher). Transfection reagents plus siRNAs/miRNAs at final concentration of 20 nM were used according to manufacturer's protocols. Cell viability was measured 72 h after transfection. ERRFI1 silencing was achieved using SMARTpool ON-TARGET plus siRNA (Dharmacon, Lafayette, CO, USA). Pre-miR™ miRNA Precursor for miR-205-5p (#PM11015) over-expression studies was from Thermo Fisher. Anti-miR-205-5p (#4101508-001) miRCURY LNA™ microRNA inhibitor was from Exiqon (Vedbaek, Denmark). Lentiviruses were produced as described in Vigna and Naldini (2000). Cells were transduced with virus titers corresponding to 40 ng/ml p24.

## mRNA, miRNA, and genomic DNA studies

Total RNA from cultured cells was extracted using miRNeasy extraction kit (Qiagen, Venlo, the Netherlands). For RNA extraction from FFPE, Maxwell® RSC RNA FFPE Kit (Promega) was used according to manufacturer's protocol, starting from 10-μm-thick sections. Retrotranscription and real-time PCR were performed as in Migliore et al (2008). Genomic DNA was extracted using QIAamp DNA Mini Kit (Qiagen). Fifty ng of gDNA was amplified and analyzed using TaqMan Gene Expression Master Mix (Thermo Fisher).

EGFR exons from 17 to 28 were sequenced using the following primers: amplicon 1 fw1: 5′-CTCCTCTTGCTGCTGGTGGT-3′ rev1: 5′-ATCTTGACATGCTGCGGTGT-3′; amplicon 2 fw2: 5′-AAAGGGCATGAACTACTTGGAG-3′ rev2: 5′-ATGAGGTACTCGTCGGCATC-3′ amplicon 3 fw3: 5′-AGAATGCATTTGCCAAGTCCTAC-3′ rev3: 5′-GCTGGACAGTGTTGAGATACTCG-3′; amplicon 4 fw4: 5′-TGCCTGAATACATAAACCAGTCC-3′, rev4: 5′-TCTGTGGGTCTAAGAGCTAATGC-3′). Mutational analysis was performed via PCR amplification of 2 μl of cDNA using AmpliTaq Gold kit (Promega). PCR products were purified using AMPure (Agencourt Bioscience Corp., Beckman Coulter S.p.A, Milan, Italy) according to manufacturer's procedures and analyzed with a 3730 DNA Analyzer, ABI capillary electrophoresis system (Thermo Fisher).

MiRNA expression profiling was performed using TaqMan Low Density Arrays (TLDA) (Array A) following manufacturer's instructions (Thermo Fisher). Each sample was run in biological duplicate. For each sample, 377 miRNAs were profiled and 98 miRNAs were selected; miRNAs showing anomalies (flag) during amplification and/or a CT value higher than 32 (indicating a low level of expression) in at least two out of the three cell lines analyzed were excluded from analysis. For miRNA analysis, expression data from each array were first normalized to the internal control RNU48; subsequently, the ΔΔCT method was used to evaluate the relative expression level (fold change) of miRNAs in each resistant subline versus the respective parental cell line. Delta-ct values were loaded in GEDAS software (Fu & Medico, 2007) to perform hierarchical clustering analysis and represent data in a heatmap.

## Methylation analysis

Sub-confluent EBC-1 and SG16 cells were treated with 5-Aza-2′-deoxycytidine (A3656- Sigma, Darmstadt, Germany) at a concentration of 1 μM for 72 h and 0.5 μM for 48 h, respectively.

For CpG methylation analysis at the miR-205 locus, 600 ng of genomic DNA was subjected to bisulfite conversion using EpiTect Bisulfite Kit (Qiagen) and subsequently PCR-amplified using the forward primer 5′-TGGAGTGAAGTTTAGGAGGTATGG-3′ and biotinylated reverse primer 5′-CACACTCCAAATATCTCCTTCATT-3′. The PCR mix contained 3 mM $MgCl_2$, 0.4 mM deoxynucleosidetriphosphates

## The paper explained

### Problem

Genomics-based precision cancer medicine is predicated upon the notion that (a) genetic alterations of so-called *driver oncogenes* are necessary to sustain cancer growth and (b) pharmacological targeting of driver oncogenes is deleterious to cancer growth. This has led to the successful development and clinical approval of "smart drugs" targeted to driver oncogenes. A major class of these "smart drugs" is represented by oncogenic receptor tyrosine kinase inhibitors (referred to as TKIs). However, the possibility of achieving long-lasting responses to TKI treatment in the clinic is often limited by the occurrence of drug resistance, a process whereby cancer cells escape from inhibition of their driver oncogenic receptor tyrosine kinase (RTK) which they depend on. Understanding ways and means through which cancer cells become refractory to TKIs is vital to circumvent resistance in the clinic.

### Results

The MET RTK has been nominated a driver oncogene in several tumor types, including lung and gastric carcinomas, with MET-TKIs being in advanced clinical development. Here, we report the discovery of a mechanism through which cancer cells become resistant to a number of structurally unrelated MET-TKIs. Specifically, we show that resistant cells upregulate the expression of miR-205, a non-coding RNA that inhibits the expression of a gene named *ERRFI1*. A known function of the ERRFI1 gene product is that of inhibiting the activity of the epidermal growth factor receptor (EGFR) RTK. Thus, the miR-205-mediated reduction in ERRFI1 expression ends up increasing EGFR activity, which drives the activation of a bypass signaling pathway, allowing MET-TKI-targeted cells to elude MET inhibition. Experimentally, this mechanism of resistance could be reverted by a combination therapy with MET and EGFR TKIs.

### Impact

Our work suggests that miR-205 upregulation and attendant loss of ERRFI1 expression flags the occurrence of an EGFR-driven mechanism of secondary resistance to MET-TKIs. Several clinically approved EGFR TKIs are already available; thus, our work has uncovered a therapeutically actionable mechanism of adaptive resistance to MET-TKIs.

dNTPs, 1 U of Platinum Taq DNA Polymerase (Invitrogen), 0.6 µM forward primer, 0.4 µM biotinylated reverse primer, and 20 ng of bisulfite-treated DNA in a final volume of 25 µl. PCR was carried out for a total of 50 cycles (30 s at 94°C, 30 s at 55°C, and 30 s at 72°C) in a PCR System 9700 (Applied Biosystems). Pyrosequencing analysis was performed according to the manufacturer's instructions using PyroMark Gold Q96 Reagents (Qiagen) and the sequencing primer 5′-GAGTTGATAATTATGAGGTT-3′ (0.5 µM in 40 µl of annealing buffer).

### Immunohistochemistry

Immunohistochemistry was performed on FFPE tissue sections using the Dako Autostainer link platform. Deparaffinization, rehydration, and target retrieval were performed in the PT Link (Dako PT100). Slides were then processed on the Autostainer Link 48 (Dako AS480) using an automated EnVision FLEX (DAKO) staining protocol. The anti-ERRFI1 mAb clone E2 was used at 5 µg/ml (Frosi

*et al*, 2010). Positive and negative controls were included for each immunohistochemical run. Pictures were acquired with the Leica Assistant Suit (LAS EZ) Software.

### RNA-Seq

Total RNA from crizotinib-resistant and parental GTL16 and EBC-1 cells was subjected to high-throughput sequencing for long poly-A+ RNAs. Two biological replicates from each cell line were analyzed. RNA sequencing was performed in an Illumina NextSeq500 sequencer, obtaining a mean of 75 million 75 bps-single reads per sample, with a stranded protocol. The Bowtie program (Langmead *et al*, 2009) was used to align reads to the reference human hg19 genome. Annotations provided by Ensembl GRCh37 and GENE-CODE 19 were set as reference for the RSEM computational pipeline (Li & Dewey, 2011) used for quantification of gene expression levels. The EBseqtool (Leng *et al*, 2013) was used to evaluate modulated genes, with | FC | > 2 and FDR < 0.1 as parameters to define the statistical significance of differential gene expression. Targets of hsa-miR-205 were extracted from the TargetScan database, version 7.1 (Agarwal *et al*, 2015).

### Data availability

The RNA-Seq data from this publication have been deposited to the GEO database (https://www.ncbi.nlm.nih.gov/geo/) and assigned the accession identifier GSE114406 (https://www.ncbi.nlm.nih.gov/geo/query/acc.cgi?acc=GSE114406). Raw sequence data are available throughout the NCBI Sequence Read Archive (https://www.ncbi.nlm.nih.gov/sra), with accession number: PRJNA451125 (https://www.ncbi.nlm.nih.gov/bioproject/PRJNA451125/)[§].

### Statistical analyses

Statistical analyses were performed with the GraphPad Prism software 7.02. One-way ANOVA or two-way ANOVA was followed by Bonferroni's multiple comparisons test as indicated in figure legends. No sample was excluded from the analysis. Graphs represent the mean, with error bars showing the SD. Shapiro–Wilk test was used to assess normality of data distributions. Statistical significance: $*P < 0.05$; $**P < 0.01$; $***P < 0.001$. All *P*-values for main figures, tables, Expanded View figures and Appendix supplementary figures can be found in Appendix Figs S11–S13.

**Expanded View** for this article is available online.

### Acknowledgements

We thank all our colleagues for helpful scientific discussion; Daniela Conticelli for help in experiment execution; Barbara Martinoglio, Roberta Porporato, Daniela Cantarella, Michela Buscarino, Stefania Bolla, and Ivana Sarotto for providing technical support with real-time PCR, RNA extraction, Sanger sequencing, and IHC, respectively; Stefania Durando and Daniel Moya for animal care; we are indebted to S. Di Agostino for advice, L. Nardinocchi for cloning the Luc-ERRFI1 reporter and to Dr. Dragani for help for methylation

[§]Correction added on 7 September 2018 after first online publication: the accession ID of the raw sequence data has been updated.

experiments. JNJ-38877605 was kindly provided by Janssen. This work was funded by the Italian Association for Cancer Research (AIRC); IG grants 20210 to SG, 16726 to OS and 20176 to AC.

## Author contributions

CM, AC, OS, and SG conceived the study; SG and CM designed the experiments; EG, EM, CM, VPL, and SA performed experiments; MA performed *in vivo* experiments; FDB and FP contributed patients' samples; AS performed pathological analyses; SG, OS, and CM wrote the manuscript; DC contributed in the RNA-Seq analysis. All authors revised the manuscript.

## Conflict of interest

The authors declare that they have no conflict of interest.

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
