## [Review Process File · EMBO Molecular Medicine]

miR-205 mediates adaptive resistance to MET inhibition via ERFF1 targeting and raised EGFR signaling

Cristina Migliore, Elena Morando, Elena Ghiso, Sergio Anastasi, Vera P. Leoni, Maria Apicella, Davide Cora', Anna Sapino, Filippo Pietrantonio, Filippo De Braud, Amedeo Columbano, Oreste Segatto and Silvia Giordano

Review timeline:

Submission date:	05 December 2017
Editorial Decision:	08 February 2018
Revision received:	17 May 2018
Editorial Decision:	14 June 2018
Revision received:	25 June 2018
Accepted:	27 June 2018

Editor: Céline Carret

Transaction Report:

1st Editorial Decision

08 February 2018

Thank you for the submission of your manuscript to EMBO Molecular Medicine and please accept my apologies for the unusual delay in getting back to you. We have now finally heard back from the three referees whom we asked to evaluate your manuscript.

You will see from the set of comments pasted below that referees 1 and 2 are supportive and only mention minor points, while referee 3 is much more critical. As EMBO Molecular Medicine focuses on studies of general interest with translational value, we agreed with some of the critical points of referee 3. Furthermore, upon our cross-commenting exercise, referees went back and forth and agreed with the following items that must be addressed for the paper to be further considered:

- 1- One way of showing if the drug is hitting the same target is demonstrating cross-resistance across the 3 compounds: therefore including the extra chemicals in all the cell lines is required (ref3, point 1)
- 2- Ref. 3 wants to see the western blot of the drug-selected cells in the absence of drug to check whether the resistant cells have not undergone signalling rewiring such that at baseline there is elevated phosphorylation of AKT and ERK (ref. 3 point 1 Fig. 2G)
- 3- authors have to show that methylation of the promoter is altering the expression of miR-205, it's only correlative now (ref. 3 point 2)
- 4- More cells should be included to validate that the effect is wide spread (or not) (ref. 3 point 3)
- 5- The full dataset for the gene expression analysis needs to be provided.
- 6- Evidence to support the findings in vivo. 2 referees out 3 support this.

We would therefore welcome the submission of a revised version within three months for further consideration and would like to encourage you to address all the criticisms raised as suggested to improve conclusiveness and clarity. Please note that EMBO Molecular Medicine strongly supports a single round of revision and that, as acceptance or rejection of the manuscript will depend on another round of review, your responses should be as complete as possible.

I look forward to receiving your revised manuscript.

***** Reviewer's comments *****

Referee #1 (Comments on Novelty/Model System for Author):

human data

Referee #1 (Remarks for Author):

The manuscript by Migliore and colleagues reports on an interesting observation that in a clinical context the miR-205 and ERRFI1 are directly involved in the resistance to MET TKI. The patient data are supported also by in vitro experiments in three cell lines. I think this is a manuscript of interest for the reader of the journal, and I have only minor comments, as this is a well-done study. Also, I consider that the identification of a patient with abnormal miR-205 and ERRFI1 levels and the expected phenotype, explain why in vivo study in mice are not necessary for this study in order to be published in the journal

Minor comments:

1 what are the Gene Ontology data for the array results presented in Figure 2H for each of the cell lines? Are these common between the cell lines?

2 what are the exact CpG sites methylated? A clear map of these loci is useful.

3 the figure 2G is nicely done and the Westerns are clear. Was in fact used only one time the normaliser, meaning all the ten hybridizations were done on only one blot? Otherwise, each normaliser for each membrane has to be presented.

3 the figure 2G is nicely done and the Westerns are clear. Was in fact used only one time the normaliser, meaning all the ten hybridizations were done on only one blot? Otherwise, each normaliser for each membrane has to be presented.

Referee #2 (Remarks for Author):

The manuscript under consideration describes a model by which, miR-205 upregulation drives resistance to MET inhibitors through a by-pass pathway involving activation of EGFR. In addition to identifying that miR-205 is upregulated, the authors determined at least one potential mechanism that miR-205 is elevated, which entails reduced promoter methylation. The authors also determined that the effect is mainly mediated through miR-205 targeting ERRFI, a negative regulator of EGFR. The manuscript concludes with validation that miR-205 and ERRFI are anti-correlated in one human tumor sample and that miR-205 is elevated, and ERRFI reduced, following resistance. Overall the study is well-controlled, the conclusions are justified and the findings are novel. The manuscript will likely be of interest to a broad audience. While the manuscript is generally acceptable, there is one figure/region of text that needs editing and perhaps an additional experiment done to support the conclusions.

The authors state that "deletion of the predicted miR-205 seed sequenceeliminated luciferase downregulation by ectopic miR-205, pointing to a direct regulation of ERRFI1 by miR-205". There are multiple issues with this statement. 1) The authors did not delete the "seed sequence" of miR-205. Seed sequence refers to the miRNA, not the target. They deleted the targeting sequence in ERRFI1. 2) Importantly, they did not delete the sequence in ERRFI1 targeted by the miR-205 seed sequence either. This would have been preferred and would have been more convincing of a direct role for miR-205 in regulating ERRFI1. Instead they deleted over 200nt of the 3' UTR of ERRFI1, which contains the miR-205 target sequence, and likely many other regulatory sequences. MiR-205 could alter the expression of other factors that bind this region generating an indirect effect. This is a rather large deletion to make the above conclusion. To conclude that miR-205 is directly targeting ERRFI1 in the predicted targeting sequence, the authors would need to mutate only a portion of the targeting sequence and not delete 200nt. 3) The authors state that they "eliminated" luciferase downregulation. This is somewhat inaccurate. Although not significant, there is still some reduction in luciferase activity, which suggest that there may be another region of the 3' UTR involved. Indeed, the majority of the effect is lost when this region of the 3' UTR is deleted, but not the full effect. The text should be edited.

Referee #3 (Comments on Novelty/Model System for Author):

A larger cell line panel and in vivo experiments are required

Referee #3 (Remarks for Author):

In this manuscript, Migliore et al. describe miR-205 mediated acquired resistance to MET inhibitors. miRNA sequencing showed upregulation of miR-205, and RNA-seq showed downregulation of ERRFI1. Genetic knockdown and expression experiments provide evidence that the upregulation of miR-205 and downregulation of ERRFI1 has a functional impact on EGFR activation, which the authors show has increased activity in the resistant cell lines. The authors suggest this enables resistance to MET-TKIs. In its current form, the data is preliminary and the manuscript lack deep mechanistic insight and has low translational value and is unsuitable for publication.

Major points

1. Consistency in experimental conditions and missing controls.

This is an incomplete study because there is a lack of consistency in the compounds used as well as missing controls. Can the authors explain why different drugs are being used for different cell lines? Why is JNJ not being used for the EBC-1 and GTL16 cell lines? Why is PHA and Crizotinib not used in the SG16 cell line? There should be consistency in the drugs and resistant cell lines used for all the figures in the manuscript.

Figure 2G makes comparisons between WT without drug and resistant cells with drug. This is not a direct comparison. The authors need to show the resistant cells without drug to confirm that the signalling in the resistant cells is not altered by the presence of the drug. Also, why was HER2 included in this blot, this is not referred to in the text.

Experiments to show that miR-205 levels are altered by overexpression or knockdown for figures 2A-F have not been provided.

Figure 2H, the authors only highlight a subset of putative miR205 targets that are downregulated in resistant cells. The entire gene expression dataset needs to be provided in the supplemental data. In particular were there any putative targets that were upregulated in the resistant cells and if so can the authors provide a reason as to why this may be the case.

2. Lack of mechanistic insights

The manuscript has not explored the molecular mechanisms as to why the resistant cells have higher levels of miR205. They make reference that demethylation may be a reason for this (figure 1E), but no further mechanistic detail has been provided

What is the molecular mechanisms of demethylation? Is it a cause or effect of resistance? Why specifically resistant cells? A sentence "studies suggest a role for de-methylation of miR-205 genomic locus" in the discussion is insufficient. The authors need to perform more experiments establishing the link between methylation and miR205 as well as the mechanism(s) by which resistant cells regulate methylation at this locus.

Fundamental questions as to whether this demethylation is dynamic are not address, e.g. do methylation levels dynamically change in response to drug treatment or is it a function of the final acquired resistant state? Is this specific to MET-TKIs are do other TKIs also induce this effect.

3. Little translational value

An important element to consider for publication in EMBO Molecular Medicine is whether these findings are of any translational value. The authors have not definitively demonstrated this.

Many of the viability changes in figure 2 and 3 are small and it is unclear if these alterations will actually have an effect on tumour growth. It is essential that *in vivo* xenograft experiments are performed to demonstrate that these small changes actually translate into a significant decrease in tumour growth. In this day and age, this would be the minimal requirement for oncology studies.

It is unclear as to what is the biomarker that is important for establishing if miR205 is the mechanism of TKI resistance. The authors only use 3 cell lines which originate from different cancer types. Given the limited number of cell lines, the authors have not demonstrated how general the observations are. Secondly the use of a larger panel of cell lines would provide clues as to whether any specific genetic features or biomarkers can indicate sensitivity to the miR205 pathway. The two case studies they provide have contrasting ERRF1 alterations upon acquisition of MET TKI resistance and hence no conclusions can be drawn about how general is this observation and which genetic factors or biomarkers dictate sensitivity to the miR205 pathway. The authors need to increase the number of cell line models and preferably in patient derived models in order to establish biomarker and molecular determinants of miR-205 driven MET TKI resistance.

Other points

1. Fig 2C, B, and D(CRIZ) show significant but not large changes, suggesting miR-205 alone is not sufficient to fully restore resistance. Dose response curves would be more informative.

2. The authors claim in page 12 that "ERRF1 expression was clearly lower in resistant cells compared to their parental counterpart" However, this is only true in 3 out of 5 resistant cell lines shown in Suppl. Fig.S2.

3. Page 11 uses the term "ERK" in the main text in reference to Fig 2G, whereas Fig 2G uses the term "MAPK" - needs to be consistent.

4. Page 11 paragraph 2 is largely descriptive and non-specific. It states that "MET largely retained sensitivity to inhibition by MET-TKIs". This statement is non-specific and Fig 2G shows that GTL16 resistant cell lines have higher MET phosphorylation than the WT+TKI. Whilst the level of phosphorylation is reduced compared to WT, this still does not show "sensitivity to inhibition" as claimed by the authors. It also claims that MET inhibition in resistant cells "did not translate into significant suppression of ERK and AKT activation", however Fig 2G shows GTL16 R-CRIZ and R-PHA cells have reduced pAKT.

5. Figure 3D does not show a "sizeable decrease of ERRF1 protein expression" in GTL16, as claimed by the authors.

6. Figure 3F needs to state is the change from CTRL to miR-205 in the right panel is non-significant.

7. Figure 4A - the authors offer no explanation as to why the case study observation in Figure 4 is completely different to the observation in S Fig S4. What is the molecular determinant of miR-205 driven MET-TKI resistance?

1st Revision - authors' response

17 May 2018

EDITOR COMMENTS

You will see from the set of comments pasted below that referees 1 and 2 are supportive and only mention minor points, while referee 3 is much more critical. As EMBO Molecular Medicine focuses on studies of general interest with translational value, we agreed with some of the critical points of referee 3. Furthermore, upon our cross-commenting exercise, referees went back and forth and agreed with the following items that must be addressed for the paper to be further considered:

1- One way of showing if the drug is hitting the same target is demonstrating cross-resistance across the 3 compounds: therefore including the extra chemicals in all the cell lines is required (ref3, point 1)

As suggested, we performed experiments assessing cell viability after exposing both wt and resistant derivatives to escalating doses of the three anti-MET drugs used in our work (i.e. Crizotinib, PHA-665752 and JNJ-3887605). Figure 1 A,B,C in the revised manuscript shows that all the resistant derivatives were strongly cross-resistant, independently from the drug to which they were initially exposed to generate resistance.

2- Ref. 3 wants to see the western blot of the drug-selected cells in the absence of drug to check whether the resistant cells have not undergone signalling rewiring such that at baseline there is elevated phosphorylation of AKT and ERK (ref. 3 point 1 Fig. 2G)

To address this issue, resistant cells were grown in the absence of the drug to which they had been rendered resistant, and lysates analyzed by WB. As shown in Figure 2I, ERK and AKT activation was not, or only marginally affected, by drug withdrawal.

3- authors have to show that methylation of the promoter is altering the expression of miR-205, it's only correlative now (ref. 3 point 2)

We performed additional experiments in which we evaluated changes in the methylation status of the miR-205 genomic region in parallel with variations in the expression levels of miR-205. As shown in Figure 1D,E of the revised manuscript, demethylation of miR-205 genomic sequences was paralleled by a sizeable increase of miR-205 expression. This datum indicates that dynamic changes of CpG methylation in the genomic region of the miR-205 locus are mirrored by variations of miR-205 expression. This provides additional support to our model of epigenetic regulation of miR-205.

4- More cells should be included to validate that the effect is wide spread (or not) (ref. 3 point 3)

In the original work we performed our experiments in three cell lines (two established cell lines and a primary cell line), all of which addicted to MET. In the revised version we added two more MET-addicted cell lines, namely KATO II and SNU-5, both of which showed anti-correlated changes of miR-205 and ERFF1 expression upon the acquisition of resistance to MET TKIs (Figures EV1E,F, EV2 B,C,D,E). However, no increase of miR-205 was observed in two other MET-addicted cell lines (H1993 and Hs746T) rendered resistant to both crizotinib and PHA-665752 (these negative data are not included in the revised version but only presented to the reviewer. If requested, we can add them as an appendix figure). Interestingly, when we expressed miR-205 in a not-MET-addicted lung cancer cell line (A549) we observed a decrease of ERFF1 expression and concomitant up-regulation of EGFR expression/activation (Appendix Figure S8). In aggregate, these experiments suggest that the described mechanism of resistance to MET TKIs, while not ubiquitous, does play a role in several cellular models of different histological origin (i.e. lung and gastric cancer).

5- The full dataset for the gene expression analysis needs to be provided.

The full dataset has now been deposited at NCBI Sequence Read Archive, SRA (accession number SUB3868037) and at NCBI Gene Expression Omnibus, GEO (accession number GSE114406).

6- Evidence to support the findings *in vivo*. 2 referees out 3 support this.

As requested, we performed xenograft experiments by injecting GTL16 resistant cells (R-CRIZ) transduced with lentiviral (LV) stocks generated from pCDH-anti-miR-205 or the empty vector pCDH (Appendix Figure S4A). Mice were treated with crizotinib (25 mg/kg) and tumor volume was monitored for 18 days. As shown in Figure 2G, anti-miR-205 transduced cells reacquired sensitivity to crizotinib treatment, while tumors generated by control cells remained insensitive to the drug. In the mirror experiment, GTL16 cells infected with either control or miR-205 LV stocks (Appendix Figure S4B) were injected in mice. When tumors reached an approximate volume of 150 mm³, treatment with crizotinib or with vehicle was started. Figure 2H shows that tumors generated by miR-205 overexpressing cells were refractory to Crizotinib treatment, while tumors generated by control cells were highly sensitive.

We performed similar *in vivo* experiments with control and ERFF11 over-expressing GTL16 R-CRIZ cells. As shown in Figure EV 3B, tumors generated by ERFF11 derivatives of GTL16 resistant cells were significantly smaller compared to control cells, implying that forced ERFF11 expression was able to partially revert resistance. Interestingly, the effect on tumor growth was less dramatic than that observed in anti-miR-205 derivatives (Figure 2H). This result suggests that, although ERFF11 appears to be the most critical miR-205 target in this context, other miRNA targets are likely to contribute to the development of the resistant phenotype. This point has been discussed in the text (page 8 of the manuscript).

In aggregate, the above data validate in tumor xenotransplants the model emerged from molecular genetics and pharmacological approaches in *in vitro* model systems.

We thank the editor and the referees for having encouraged us to pursue *in vivo* experiments that further strengthen the relevance of our findings.

Referee #1 (Remarks for Author):

The manuscript by Migliore and colleagues reports on an interesting observation that in a clinical context the miR-205 and ERFF11 are directly involved in the resistance to MET TKI. The patient data are supported also by in vitro experiments in three cell lines. I think this is a manuscript of interest for the reader of the journal, and I have only minor comments as this is a well-done study. Also, I consider that the identification of a patient with abnormal miR-205 and ERFF11 levels and the expected phenotype, explain why in vivo study in mice are not necessary for this study in order to be published in the journal

We thank the reviewer for the positive evaluation of our work.

Minor comments:

I what are the Gene Ontology data for the array results presented in Figure 2H for each of the cell lines? Are these common between the cell lines?

We performed the GO analysis for the genes presented in the new Figure 2L. The results show that the pathways in which downregulated miR-205 target genes are involved are not shared between the two models (see the graphs below for reviewer only). A hypothesis to explain this result is that in our NGS analysis we identified a small number of miR-205 downregulated targets and thus the GO analysis was quite restricted. Indeed, only three targets are shared by the two models, namely ERFF11, Epiregulin and ZEB1. Epiregulin downregulation could seem unexpected; however, as we observed a general rewiring of EGFR ligands, the observed Epiregulin decrease could be biologically irrelevant due to the compensatory presence of other EGFR ligands.

MiR-205 target genes downregulated in EBC-1 cells: Gene Ontology

ENSG00000001561_ENPP4	ENPP4
ENSG00000148516_ZEB1	ZEB1
ENSG00000164124_TMEM144	TMEM144
ENSG00000185716_C16orf52	C16orf52
ENSG00000095203_EPB41L4B	EPB41L4B
ENSG00000108960_MMD	MMD
ENSG00000002587_HS3ST1	HS3ST1
ENSG00000177685_EFCAB4A	EFCAB4A
ENSG00000073969_NSF	NSF
ENSG00000088538_DOCK3	DOCK3
ENSG00000164056_SPRY1	SPRY1
ENSG00000171617_ENC1	ENC1
ENSG00000173821_RNF213	RNF213
ENSG00000106868_SUSD1	SUSD1
ENSG00000159023_EPB41	EPB41
ENSG00000116285_ERRF1	ERRF1
ENSG00000154229_PRKCA	PRKCA
ENSG00000149177_PTPRJ	PTPRJ
ENSG00000095637_SORBS1	SORBS1
ENSG00000114126_TFDP2	TFDP2
ENSG00000137819_PQR5	PAQR5
ENSG00000128656_CHN1	CHN1
ENSG00000214338_SOGA3	SOGA3
ENSG00000124882_EREG	EREG
ENSG00000169306_IL1RAPL1	IL1RAPL1
ENSG00000144827_ABHD10	ABHD10
ENSG00000138639_ARHGAP24	ARHGAP24
ENSG00000151276_MAGI1	MAGI1
ENSG00000112715_VEGFA	VEGFA
ENSG00000141279_NPEPPS	NPEPPS
ENSG00000114030_KPNA1	KPNA1
ENSG00000133318_RTN3	RTN3
ENSG00000133816_MICAL2	MICAL2
ENSG00000204396_VWA7	VWA7
ENSG00000267882_ALO31666.2	ALO31666.2
ENSG00000089818_NECAP1	NECAP1
ENSG00000134686_PHC2	PHC2
ENSG00000138623_SEMA7A	SEMA7A
ENSG00000158669_AGPAT6	AGPAT6
ENSG00000142784_WDTC1	WDTC1
ENSG00000131759_RARA	RARA
ENSG00000153395_LPCAT1	LPCAT1
ENSG00000170035_UBE2E3	UBE2E3
ENSG00000164530_PI16	PI16

**GO Cellular Component
2017b**

Golgi stack lumen (GO:0034469)

dendrite (GO:0030425)

cortical cytoskeleton (GO:0030863)

CA3 pyramidal cell dendrite (GO:0097442)

sensory dendrite (GO:0071683)

**GO Biological Process
2017b**

positive regulation of cell adhesion (GO:004

positive regulation of cell adhesion mediate

positive regulation of cell-cell adhesion (GO:

positive regulation of cell-substrate adhesio

phosphorylation of RNA polymerase II C-ter

**GO Molecular Function
2017b**

protein kinase B binding (GO:0043422)

GTPase activator activity (GO:0005096)

platelet-derived growth factor receptor bind

4-nitrophenol 2-monooxygenase activity (G

SH3/SH2 adaptor activity (GO:0005070)

<http://amp.pharm.mssm.edu/Enrichr/>**Enrichr**

Analyze What's New? Libraries Find a Gene About Help

MiR-205 target genes downregulated in GTL16 cells: Gene Ontology

ENSG00000047644_WWC3	WWC3
ENSG00000185565_LSAMP	LSAMP
ENSG00000135862_LAMC1	LAMC1
ENSG00000088854_C20orf194	C20orf194
ENSG00000102804_TSC22D1	TSC22D1
ENSG00000116285_ERRFI1	ERRFI1
ENSG00000135643_KCNMB4	KCNMB4
ENSG00000095739_BAMBI	BAMBI
ENSG00000204386_NEU1	NEU1
ENSG00000123407_HOXC12	HOXC12
ENSG00000125347_IRF1	IRF1
ENSG00000184702_SEPT5	SEPT5
ENSG00000054523_KIF1B	KIF1B
ENSG00000162849_KIF26B	KIF26B
ENSG00000163697_APBB2	APBB2
ENSG00000115977_AAK1	AAK1
ENSG00000124882_EREK	EREK
ENSG00000137801_THBS1	THBS1
ENSG00000148516_ZEB1	ZEB1
ENSG00000153250_RBMS1	RBMS1
ENSG00000059728_MXD1	MXD1
ENSG00000177189_RPS6KA3	RPS6KA3
ENSG00000117707_PROX1	PROX1
ENSG00000182985_CADM1	CADM1

**GO Cellular Component
2017b**

kinesin complex (GO:0005871)

kinesin II complex (GO:0016939)

minus-end kinesin complex (GO:0005872)

plus-end kinesin complex (GO:0005873)

kinesin I complex (GO:0016938)

**GO Biological Process
2017b**

negative regulation of transcription from RNA polymerase II promoter

positive regulation of epithelial cell proliferation

negative regulation of growth plate cartilage development

negative regulation of transcription, DNA-dependent

positive regulation of fibroblast proliferation

**GO Molecular Function
2017b**

transcriptional repressor activity, RNA polymerase II

transcription corepressor activity (GO:0003700)

RNA polymerase II transcription corepressor activity

microtubule motor activity (GO:0003777)

ATP-dependent microtubule motor activity

<http://amp.pharm.mssm.edu/Enrichr/>
Analyze What's New? Libraries Find a Gene About Help

2 what are the exact CpG sites methylated? A clear map of these loci is useful.

We apologize for not having included the map of the analyzed CpGs in the previous version. The map is now shown in Appendix Figure S2.

3 the figure 2G is nicely done and the Westerns are clear. Was in fact used only one time the normaliser, meaning all the ten hybridizations were done on only one blot? Otherwise, each normaliser for each membrane has to be presented.

We apologize for not showing the normalizer for each of the two blots in Figure 2G of the original submission. In addressing a point raised by Reviewer 3, we modified the blots shown in the actual Figure 2I and the actin normalizer is now shown for both of them.

Referee #2 (Remarks for Author):

The manuscript under consideration describes a model by which, miR-205 upregulation drives resistance to MET inhibitors through a by-pass pathway involving activation of EGFR. In addition to identifying that miR-205 is upregulated, the authors determined at least one potential mechanism that miR-205 is elevated, which entails reduced promoter methylation. The authors also determined that the effect is mainly mediated through miR-205 targeting ERRFI, a negative regulator of EGFR. The manuscript concludes with validation that miR-205 and ERRFI are anti-correlated in one human tumor sample and that miR-205 is elevated, and ERRFI reduced, following resistance. Overall the study is well-controlled, the conclusions are justified and the findings are novel. The manuscript will likely be of interest to a broad audience. While the manuscript is generally acceptable, there is one figure/region of text that needs editing and perhaps an additional experiment done to support the conclusions.

We thank the reviewer for the positive evaluation of our work.

The authors state that "deletion of the predicted miR-205 seed sequenceeliminated luciferase downregulation by ectopic miR-205, pointing to a direct regulation of ERRFI1 by miR-205". There are multiple issues with this statement. 1) The authors did not delete the "seed sequence" of miR-205. Seed sequence refers to the miRNA, not the target. They deleted the targeting sequence in ERRFI1. 2) Importantly, they did not delete the sequence in ERRFI1 targeted by the miR-205 seed sequence either. This would have been preferred and would have been more convincing of a direct role for miR-205 in regulating ERRFI1. Instead they deleted over 200nt of the 3' UTR of ERRFI1, which contains the miR-205 target sequence, and likely many other regulatory sequences. MiR-205 could alter the expression of other factors that bind this region generating an indirect effect. This is a rather large deletion to make the above conclusion. To conclude that miR-205 is directly targeting ERRFI1 in the predicted targeting sequence, the authors would need to mutate only a portion of the targeting sequence and not delete 200nt. 3) The authors state that they "eliminated" luciferase downregulation. This is somewhat inaccurate. Although not significant, there is still some reduction in luciferase activity, which suggest that there may be another region of the 3' UTR involved. Indeed, the majority of the effect is lost when this region of the 3' UTR is deleted, but not the full effect. The text should be edited.

We thank the Reviewer for pointing out our mistake. The text has been edited to comply with the Reviewer's objections. As suggested by the reviewer, we have mutagenized the miR-205 targeted sequence in the ERRFI1 3' UTR. Figure 3F shows that the luciferase activity of the mutated reporter is not affected by ectopic miR-205, which provides cogent proof that this sequence is indeed targeted by miR-205 in our cellular models.

Referee #3 (Comments on Novelty/Model System for Author):

A larger cell line panel and in vivo experiments are required

Referee #3 (Remarks for Author): In this manuscript, Migliore et al. describe miR-205 mediated acquired resistance to MET inhibitors. miRNA sequencing showed upregulation of miR-205, and RNA-seq showed downregulation of ERRFI1. Genetic knockdown and expression experiments provide evidence that the upregulation of miR-205 and downregulation of ERRFI1 has a functional impact on EGFR activation, which the authors show has increased activity in the resistant cell lines.

The authors suggest this enables resistance to MET-TKIs. In its current form, the data is preliminary and the manuscript lack deep mechanistic insight and has low translational value and is unsuitable for publication.

Major points

1. Consistency in experimental conditions and missing controls.

This is an incomplete study because there is a lack of consistency in the compounds used as well as missing controls. Can the authors explain why different drugs are being used for different cell lines? Why is JNJ not being used for the EBC-1 and GTL16 cell lines? Why is PHA and Crizotinib not used in the SG16 cell line? There should be consistency in the drugs and resistant cell lines used for all the figures in the manuscript.

As pointedly suggested by the editor, we tested cross-resistance among the different derivatives and found that they display extensive cross-resistance to all the MET kinase inhibitors used in our study (Figure 1A,B,C). This datum indicates that our cell lines developed resistance to structurally different MET TKIs which share a common mechanism of action. This datum is also compatible with a model whereby the same mechanism of resistance (i.e. signal rewiring via the miR-205-ERRFI1-EGFR circuit) allows MET addicted cells to escape from pharmacological inhibition of the MET kinase by different TKIs.

Figure 2G makes comparisons between WT without drug and resistant cells with drug. This is not a direct comparison. The authors need to show the resistant cells without drug to confirm that the signalling in the resistant cells is not altered by the presence of the drug. Also, why was HER2 included in this blot, this is not referred to in the text.

According to the referee's suggestion, we performed a WB analysis of both resistant and WT cells, either in presence or absence of MET TKIs. As shown in the new Figure 2I we did not observe major differences in the activation of downstream signaling molecules when resistant derivatives were grown in the absence of the drugs. We agree that the HER2 expression/activation status is not informative and this panel was removed from the current manuscript version.

Experiments to show that miR-205 levels are altered by overexpression or knockdown for figures 2A-F have not been provided.

We regret that we did not include the requested controls in the previous version of the paper. These data are now shown in Appendix Figure S3.

Figure 2H, the authors only highlight a subset of putative miR205 targets that are downregulated in resistant cells. The entire gene expression dataset needs to be provided in the supplemental data. In particular were there any putative targets that were upregulated in the resistant cells and if so can the authors provide a reason as to why this may be the case.

The main focus of our study was to investigate whether upregulated miRNA expression could underpin resistance. This focus led us to analyze genes showing an expression pattern opposite to that of the miRNAs of interest. However, we did find a subset of putative miR-205 targets that was upregulated, as shown in the list provided in Appendix Figure S6. In the context of resistance to MET TKIs we suspect that some putative miR-205 targets could be upregulated despite increased miR-205 expression because they are either poor miR-205 targets or liable to regulatory cues capable of overruling miR-205-dependent control.

2. Lack of mechanistic insights

The manuscript has not explored the molecular mechanisms as to why the resistant cells have higher levels of miR205. They make reference that demethylation may be a reason for this (figure 1E), but no further mechanistic detail has been provided

What is the molecular mechanisms of demethylation? Is it a cause or effect of resistance? Why specifically resistant cells? A sentence "studies suggest a role for de-methylation of miR-205 genomic locus" in the discussion is insufficient. The authors need to perform more experiments establishing the link between methylation and miR205 as well as the mechanism(s) by which resistant cells regulate methylation at this locus.

Fundamental questions as to whether this demethylation is dynamic are not address, e.g. do methylation levels dynamically change in response to drug treatment or is it a function of the final acquired resistant state? Is this specific to MET-TKIs are do other TKIs also induce this effect.

We aimed at identifying mechanism/s involved in miR-205 upregulation in our cellular models, which led us to investigate the methylation status of the miR-205 genomic region. To address some of the reviewer's criticisms, we performed additional experiments to seek correlations between dynamic changes of the miR-205 locus methylation status and miR-205 levels. As shown in Figure 1D,E, we observed that miR-205 expression was significantly increased in wt (i.e. TKI-sensitive) cell lines upon 5-Aza-2'-deoxycytidine-induced de-methylation of CpG islands mapping to the miR-205 locus. We think that these novel data strengthen our model that epigenetic modifications may underpin miR-205 regulation. We do agree that additional experimentation would be required to further solidify this model. Yet, we feel that additional in-depth work would go beyond the scope of our work and would add limited value to the prevalently translational angle of this study.

3. Little translational value

An important element to consider for publication in EMBO Molecular Medicine is whether these findings are of any translational value. The authors have not definitively demonstrated this. Many of the viability changes in figure 2 and 3 are small and it is unclear if these alterations will actually have an effect on tumour growth. It is essential that in vivo xenograft experiments are performed to demonstrate that these small changes actually translate into a significant decrease in tumour growth. In this day and age, this would be the minimal requirement for oncology studies.

We acknowledge that this is a truly critical issue and we have strived to address the Reviewer's point. We performed xenograft experiments by injecting GTL16 resistant cells (R-CRIZ) transduced with control or anti-miR-205 lentivirus stocks, (Appendix Figure S4A). Mice were treated with crizotinib and tumor volume was monitored for 18 days. As shown in Figure 2G, anti-miR-205 expression reverted resistance, while tumors generated by control cells remained insensitive to the drug. In the mirror experiment (Figure 2H), tumors generated by wt GTL16 miR-205 overexpressing cells became refractory to crizotinib treatment, while tumors generated by control cells remained sensitive.

All together these results support the hypothesis that miR-205 upregulation causes MET-addicted cancer cells to acquire resistance to MET-TKIs.

We also performed *in vivo* experiments using GTL16 R-CRIZ resistant cells expressing ectopic ERRF1. As shown in Figure EV 3B, tumors generated by GTL16 R-CRIZ-ERRF1 cells were significantly smaller compared to controls, implying that re-expression of ERRF1 was able to partially revert resistance. Interestingly, the effect on tumor growth was less dramatic that what observed in tumors generated by GTL16 pCDH-anti-miR-205 cells (Figure 2H). This result suggests that while ERRF1 is likely the most critical miR-205 target in this context, other miRNA target molecules can contribute to the resistant phenotype.

We believe that the above *in vivo* data strengthen our model and enhance its translational value. We thank the Reviewer for encouraging us to pursue this experimentation.

It is unclear as to what is the biomarker that is important for establishing if miR205 is the mechanism of TKI resistance. The authors only use 3 cell lines which originate from different cancer types. Given the limited number of cell lines, the authors have not demonstrated how general the observations are. Secondly the use of a larger panel of cell lines would provide clues as to whether any specific genetic features or biomarkers can indicate sensitivity to the miR205 pathway. The two case studies they provide have contrasting ERRF1 alterations upon acquisition of MET TKI resistance and hence no conclusions can be drawn about how general is this observation and which genetic factors or biomarkers dictate sensitivity to the miR205 pathway. The authors need to increase the number of cell line models and preferably in patient derived models in order to establish biomarker and molecular determinants of miR-205 driven MET TKI resistance.

We agree that the diagnostic/prognostic value of a biomarker gains robustness when the biomarker is validated across different models. It is also true, however, that different mechanisms of resistance

can develop within each tumor type, an undeniable complexity that poses a serious, and yet ineludible, challenge to biomarker discovery/development. This is indeed the case when resistance does not have a genetic basis. Here we provide evidence that in at least three different cellular contexts of MET addiction (lung adenocarcinoma –EBC-1; gastric carcinoma – GTL16; primary gastric cells – SG16) resistance to MET TKIs may be caused by deregulation of the miR-205-ERRFI1-EGFR axis.

As requested, we investigated the other MET-addicted cellular models in our hands to verify if this mechanism could be even more general. We generated KATO-II and SNU-5 cells which showed increased levels of miR-205 and decreased expression of ERRFI1 (Figure EV1E,F, Figure EV2B,C,D,E). These results provide further support that miR-205 is one of the mediators of acquired resistance to MET TKIs.

On the other hand, when we rendered H1993 (lung cancer) and Hs-746t (gastric cancer) cells resistant to MET TKIs, we found that miR-205 was not expressed in either wt or resistant derivatives (see table below reporting qPCR analysis), implying that other mechanism/s underpin MET TKIs resistance in these cells.

As for biomarker development, we believe that this goes beyond the scope of our work. The possibility of using ERRFI1 and miR-205 levels as predictive biomarkers is elaborated in the Discussion.

	miR-205	RNU48
H1993 WT	Undetermined	19,605
H1993 WT	Undetermined	19,628
H1993 WT	38,202	19,752
H1993 R200 CRIZ	37,936	20,494
H1993 R200 CRIZ	Undetermined	20,181
H1993 R200 CRIZ	Undetermined	20,748
H1993 R800 CRIZ	37,458	22,364
H1993 R800 CRIZ	36,526	22,126
H1993 R800 CRIZ	37,125	22,259
H1993 R150 PHA	39,753	19,413
H1993 R150 PHA	37,947	19,630
H1993 R150 PHA	Undetermined	19,648
	miR-205	RNU48
HS746T WT	35,002	19,892
HS746T WT	35,760	19,908
HS746T WT	36,702	19,763
HS746T R250 CRIZ	Undetermined	21,567
HS746T R250 CRIZ	Undetermined	21,729
HS746T R250 CRIZ	Undetermined	21,712
HS746T R600 PHA	38,114	25,054
HS746T R600 PHA	Undetermined	25,157
HS746T R600 PHA	Undetermined	25,094

We also verified if miR-205 overexpression could be responsible for resistance to other TKIs. We looked at miR-205 expression in HCC827, HCC4006, H3255 and PC-9 EGFR-addicted lung cancer cell lines rendered resistant to either Erlotinib, Gefitinib, Afatinib, AZD8931 or Dacomitinib. In none of these twenty models we observed a consistent modulation of miR-205. Only in HCC827 cells resistant to Gefitinib we observed a 4 fold increase of miR-205, which is far below the average increase observed in MET-addicted cells we studied (Appendix Figure S10).

As already stated in the main text, samples from tumors addicted to MET activation which develop resistance to MET TKIs are extremely rare because MET targeted treatments have not been approved yet. Thus, validation of our model on a larger scale of clinical samples is not feasible for the time being. That said, we believe that the validation in a patient of the mechanism of resistance we discovered *in vitro* provides an initial but highly encouraging proof of concept of our model. While we are not in a position to comment on how general this mechanism might be, we see that the combination of *in vivo* experimentation reported in Figure 2 G,H of the revised manuscript and clinical data in Figure 4 provides strong translational value. Finally, the fact that two patients experienced different mechanisms of resistance is in agreement with the consolidated notion that targeted treatment may generate different mechanisms of resistance.

Other points

1. Fig 2C, B, and D(CRIZ) show significant but not large changes, suggesting miR-205 alone is not sufficient to fully restore resistance. Dose response curves would be more informative.

We agree with the reviewer that the differences are not always impressive. We note that the largest differences were observed when miR-205 expression was inhibited in cells with the highest miR-205 expression (SG16 and EBC-1 resistant derivatives) or when miR-205 was overexpressed in cells with a low endogenous amount (wt GTL16 cells).

2. The authors claim in page 12 that "ERRF1 expression was clearly lower in resistant cells compared to their parental counterpart" However, this is only true in 3 out of 5 resistant cell lines shown in Suppl. Fig.S2.

We have quantified the intensity of the WB bands shown in Figure EV 2A. The graph clearly demonstrates that 4/5 resistant derivatives show decreased ERRF1 expression. This was not the case of GTL16 R-PHA that, as stated in the text, relied for resistance on KRAS amplification and not on ERRF1 downregulation/EGFR activation. We also note that relatively minor changes in ERRF1 expression have been reported to have a strong impact on EGFR activation in *in vitro* and *in vivo* model systems (Anastasi et al., 2016).

3. Page 11 uses the term "ERK" in the main text in reference to Fig 2G, whereas Fig 2G uses the term "MAPK" - needs to be consistent.

We apologize for that discrepancy. In the revised version we consistently used the term ERK all over the manuscript.

4. Page 11 paragraph 2 is largely descriptive and non-specific. It states that "MET largely retained sensitivity to inhibition by MET-TKIs". This statement is non-specific and Fig 2G shows that GTL16 resistant cell lines have higher MET phosphorylation than the WT+TKI. Whilst the level of phosphorylation is reduced compared to WT, this still does not show "sensitivity to inhibition" as claimed by the authors. It also claims that MET inhibition in resistant cells "did not translate into significant suppression of ERK and AKT activation", however Fig 2G shows GTL16 R-CRIZ and R-PHA cells have reduced pAKT.

We agree with the reviewer that the level of MET activation in resistant cells is higher than that observed in wt cells exposed to TKIs. However, the partial sensitivity of resistant cells to MET TKIs is testified by the observed increase in MET phosphorylation upon TKI removal (see Figure 2I). Furthermore, because GTL16 are *MET* amplified, it is possible that residual MET Tyr phosphorylation upon MET TKI treatment reflects cross phosphorylation by EGFR. Concerning GTL16 resistant cells, it is true that at baseline they show pAKT levels lower than wt cells; however AKT activation is clearly higher than in wt cells exposed to the MET TKI, which, in several models, has been shown to be sufficient to generate resistance. We have edited the text to take into account the above details and offer a more precise and balanced presentation of data in Figure 2I.

5. Figure 3D does not show a "sizeable decrease of ERRF1 protein expression" in GTL16, as claimed by the authors.

We agree that ERRF1 decrease is not homogeneous in all the cell lines (ranging from 50 to 80%, as shown by the added quantitation – Figure 3D). We have changed the sentence by removing “sizeable”.

6. *Figure 3F needs to state is the change from CTRL to miR-205 in the right panel is non-significant.*

We apologize for having forgotten to show that the difference was not significant. This has now been added in the figure.

7. *Figure 4A - the authors offer no explanation as to why the case study observation in Figure 4 is completely different to the observation in S Fig S4. What is the molecular determinant of miR-205 driven MET-TKI resistance?*

As stated, patient #1 became resistant to MET TKI as a consequence of further MET amplification (Oddo et al, BJC, 117:347-352, 2017); no change in the miR-205/ERRF1/EGFR axis was observed (Figure EV4). Patient's #2 tumor, instead, showed increased miR-205 expression and ERRF1 downregulation (Figure 4), which we pinpointed as a mechanism of clinical resistance to MET TKIs by analogy with our *in vitro* and *in vivo* experimental models. While we acknowledge that additional molecular analyses would be required to provide further support to our model in the clinical setting, we note that the scarcity of pathological material obtained at biopsy precluded further studies.

2nd Editorial Decision

14 June 2018

Thank you for the submission of your revised manuscript to EMBO Molecular Medicine. We have now received the enclosed reports from the referees that were asked to re-assess it. As you will see the reviewers are now supportive and I am pleased to inform you that we will be able to accept your manuscript pending final editorial amendments.

Please submit your revised manuscript within two weeks. I look forward to seeing a revised form of your manuscript as soon as possible.

***** Reviewer's comments *****

Referee #2 (Comments on Novelty/Model System for Author):

none

Referee #2 (Remarks for Author):

The authors addressed all of the previous critiques appropriately.

Referee #3 (Remarks for Author):

The authors have addressed all my concerns in this revision and this manuscript is now suitable for EMBO Molecular Medicine.

Corresponding Author Name: SILVIA GIORDANO, ORESTE SEGATTO, CRISTINA MIGLIORE

Journal Submitted to: EMBO MOLECULAR MEDICINE

Manuscript Number: EMM-2017-08746-V2